# Three Towers: Flexible Contrastive Learning with Pretrained Image Models

**Jannik Kossen**[1,∇,△]   **Mark Collier**[2,∇]   **Basil Mustafa**[3]   **Xiao Wang**[3]   **Xiaohua Zhai**[3]

**Lucas Beyer**[3]   **Andreas Steiner**[3]   **Jesse Berent**[2]   **Rodolphe Jenatton**[3,□]   **Efi Kokiopoulou**[2,□]

[1] OATML, Department of Computer Science, University of Oxford
[2] Google Research    [3] Google DeepMind

## Abstract

We introduce Three Towers (3T), a flexible method to improve the contrastive learning of vision-language models by incorporating pretrained image classifiers. While contrastive models are usually trained from scratch, LiT [85] has recently shown performance gains from using pretrained classifier embeddings. However, LiT directly replaces the image tower with the frozen embeddings, excluding any potential benefits from training the image tower contrastively. With 3T, we propose a more flexible strategy that allows the image tower to benefit from both pretrained embeddings and contrastive training. To achieve this, we introduce a third tower that contains the frozen pretrained embeddings, and we encourage alignment between this third tower and the main image-text towers. Empirically, 3T consistently improves over LiT and the CLIP-style from-scratch baseline for retrieval tasks. For classification, 3T reliably improves over the from-scratch baseline, and while it underperforms relative to LiT for JFT-pretrained models, it outperforms LiT for ImageNet-21k and Places365 pretraining.

## 1   Introduction

Approaches such as CLIP [58] and ALIGN [34] have popularized the contrastive learning of aligned image and text representations from large scale web-scraped datasets of image-caption pairs. Compared to image-only contrastive learning, e.g. [51, 9, 25], the bi-modal image-text objective allows these approaches to perform tasks that require language understanding, such as retrieval or zero-shot classification [42, 58, 34]. Compared to traditional transfer learning from supervised image representations [52, 68, 44, 38], contrastive approaches can forego expensive labelling and instead collect much larger datasets via inexpensive web-scraping [58, 11, 64]. A growing body of work seeks to improve upon various aspects of contrastive vision-language modelling, cf. related work in §5.

CLIP and ALIGN train the image and text towers from randomly initialized weights, i.e. 'from scratch'. However, strong pretrained models for either image or text inputs are often readily available, and one may benefit from their use in contrastive learning. Recently, Zhai et al. [85] have shown that pretrained classifiers can be used to improve downstream task performance. They propose LiT, short for 'locked-image text tuning', which is a variation of the standard CLIP/ALIGN setup that uses frozen embeddings from a pretrained classifier as the image tower. In other words, the text tower in LiT is contrastively trained from scratch to match locked and pretrained embeddings in the image tower. Incorporating knowledge from pretrained models into contrastive learning is an important research direction, and LiT provides a simple and effective recipe for doing so.

However, a concern with LiT is that it may be overly reliant on the pretrained model, completely missing out on any potential benefits the image tower might get from contrastive training. Zhai et al.

---

[∇]Equal contribution. [△] Work done while interning at Google.   [□] Equal advising.
Correspondence to jannik.kossen@cs.ox.ac.uk and markcollier@google.com.

37th Conference on Neural Information Processing Systems (NeurIPS 2023).

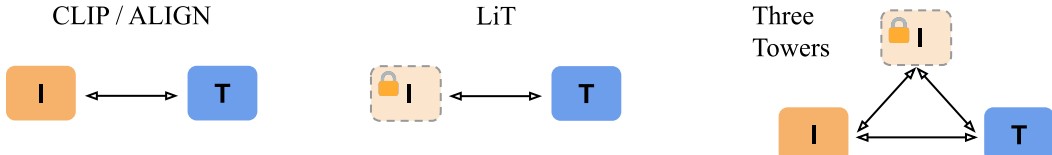

Figure 1: CLIP and ALIGN do not make use of pretrained models, and LiT directly uses a frozen pretrained model as the image tower. With Three Towers (3T), we propose a flexible strategy to improve contrastive learning with pretrained models: in addition to a pair of CLIP-style from-scratch image and text towers, we introduce a third tower which contains fixed pretrained image embeddings; extra loss terms align the main towers to the third tower. Unlike for CLIP/ALIGN and LiT, the image tower can benefit from both contrastive learning and pretrained classifier embeddings.

[85] themselves give one example where LiT performs worse than standard contrastive training: when using models pretrained on Places365 [87]—a dataset relating images to the place they were taken—the fixed embeddings do not generalize to downstream tasks such as ImageNet-1k (IN-1k) [40, 61] or CIFAR-100 [40]. For their main results, Zhai et al. [83] therefore use models pretrained on datasets such as ImageNet-21k (IN-21k) [14, 61] and JFT [68, 84] which cover a variety of classes and inputs. However, even then, we believe that constraining the image tower to fixed classifier embeddings is not ideal: later, we will show examples where LiT performs worse than standard contrastive learning due to labels or input examples not covered by IN-21k, cf. §4.2. Given the scale and variety of contrastive learning datasets, we believe it should be possible to improve the image tower by making use of both pretrained models *and* contrastive training.

In this work, we propose Three Towers (3T): a flexible approach that improves the contrastive learning of vision-language models by effectively transferring knowledge from pretrained classifiers. Instead of locking the main image tower, we introduce a third tower that contains the embeddings of a frozen pretrained model. The main image and text towers are trained from scratch and aligned to the third tower with additional contrastive loss terms (cf. Fig. 1). Only the main two towers are used for downstream task applications such that no additional inference costs are incurred compared to LiT or a CLIP/ALIGN baseline. This simple approach allows us to explicitly trade off the

Table 1: For retrieval, 3T improves on LiT and the CLIP-style baseline (top-1 recall ↑). Models are g scale, using Text-Filtered WebLI, and JFT pretraining for LiT/3T, cf. §4.1.

| Method | Basel. | LiT | 3T |
| --- | --- | --- | --- |
| Flickr img2txt | 85.0 | 83.9 | 87.3 |
| Flickr txt2img | 67.0 | 66.5 | 72.1 |
| COCO img2txt | 60.0 | 59.5 | 64.1 |
| COCO txt2img | 44.7 | 43.6 | 48.5 |

main contrastive learning objective against the transfer of prior knowledge from the pretrained model. Compared to LiT, the image tower in 3T can benefit from both contrastive training *and* the pretrained model. We highlight the following methodological and empirical contributions:

- We propose and formalize the 3T method for flexible and effective transfer of pretrained classifiers into contrastive vision-language models (§3).
- 3T consistently improves over LiT and a from-scratch baseline for retrieval tasks (e.g. Table 1, §4.1).
- For classification tasks, 3T outperforms LiT and the baseline with IN-21k pretrained models; for JFT pretraining, 3T outperforms the baseline but not LiT (§4.2).
- We extend the evaluation of Zhai et al. [85] to additional tasks and pretraining datasets, showing that 3T is significantly more robust than LiT to deficits in the pretrained model (§4.2 and §4.4).
- We show that 3T benefits more from model size or training budget increases than LiT (§4.3).
- We introduce a simple post-hoc method that allows us to further improve performance by combining 3T- and LiT-like prediction (§4.5).

## 2 Background: Contrastive Learning of Vision-Language Models

Before introducing 3T, we recap contrastive learning of vision-language models as popularized by [58, 34, 86] and give a more formal introduction to LiT [85].

**CLIP/ALIGN.** We assume two parameterized models: an image tower $f = f_\theta$ with parameters $\theta$ and a text tower $g = g_\phi$ with parameters $\phi$. Each input sample $(I_i, T_i)$ consists of a pair of matching image $I_i \in \mathcal{I}$ and text $T_i \in \mathcal{T}$, and the contrastive loss is computed over a batch $i \in \{1, \ldots, N\}$

of examples. The towers map the input modalities to a common $D$-dimensional embedding space, $f : \mathcal{I} \to \mathbb{R}^D$ and $g : \mathcal{T} \to \mathbb{R}^D$. We further assume that $f$ and $g$ produce embeddings that are normalized with respect to their L2 norm, $\|f(I)\|_2 = \|g(T)\|_2 = 1$ for any $I \in \mathcal{I}$ and $T \in \mathcal{T}$. For a batch of input samples, the bi-directional contrastive loss [66, 77, 51, 9, 86] is computed as

$$\mathcal{L}_{f \leftrightarrow g} = \frac{1}{2}(\mathcal{L}_{f \to g} + \mathcal{L}_{g \to f}), \text{ where} \tag{1}$$

$$\mathcal{L}_{f \to g} = -\frac{1}{N} \sum_{i=1}^{N} \log \frac{\exp(f(I_i)^\top g(T_i) / \tau)}{\sum_{j=1}^{N} \exp(f(I_i)^\top g(T_j)/\tau)}, \tag{2}$$

$$\mathcal{L}_{g \to f} = -\frac{1}{N} \sum_{i=1}^{N} \log \frac{\exp(f(I_i)^\top g(T_i)/\tau)}{\sum_{j=1}^{N} \exp(f(I_j)^\top g(T_i)/\tau)}. \tag{3}$$

Here, $\tau$ is a learned temperature parameter and $f(I)^\top g(T) \in \mathbb{R}$ are dot products. The two directional loss terms, $\mathcal{L}_{f \to g}$ and $\mathcal{L}_{g \to f}$, have a natural interpretation as standard cross-entropy objectives for classifying the correct matches in each batch. The parameters $\theta$ and $\phi$ of the two towers, $f_\theta$ and $g_\phi$, are jointly updated with standard stochastic optimization based on Eq. (1).

**Downstream Tasks.** After training, $f$ and $g$ are treated as fixed representation extractors. For retrieval, the dot product $f(I)^\top g(T) \in \mathbb{R}$ ranks similarity between inputs. For few-shot image classification, a linear classifier is trained atop the feature representations of $f$ from few examples; $g$ is not used. For zero-shot image classification, $f$ embeds images and $g$ all possible class labels (see [58]). For each image, one predicts the label with the largest dot product similarity in embedding space.

**LiT.** Zhai et al. [85] initialize the parameters $\theta$ of the image tower from a pretrained classifier and then keep them frozen them during training. That is, only the parameters $\phi$ of the text tower are optimized during contrastive training. As the image tower $f_\theta$, LiT uses the pre-softmax embeddings of large scale classifiers, such as vision transformers [17] trained on JFT-3B [84] or IN-21k. During contrastive training, the text tower is trained from scratch using the same objective Eq. (1).

Experimentally, Zhai et al. [85] investigate all combinations for 'training from scratch', locking and finetuning a pretrained model for both towers on a custom union of the CLIP-subset of YFCC-100M [69] and CC12M [7] (cf. Fig. 3 in [85]). For the image tower, locking gives a significant lead on IN-1k over finetuning and training from scratch, and performs similarly to finetuning and better than training from scratch for retrieval. For the text tower, a locked configuration performs badly, and finetuning gives small to negligible gains over training from scratch. Given these results, Zhai et al. [85] choose the 'locked image tower and from-scratch text tower' setup that they call LiT. At large scale, they show that a locked image tower outperforms from-scratch training and finetuning on zero-shot IN-1k, ImageNet-v2 (IN-v2) [59], CIFAR-100, and Oxford-IIIT Pet [53] classification tasks. They further show LiT outperforms CLIP/ALIGN on IN-R [31], IN-A [32], and ObjectNet [3].

While Zhai et al. [85] show strong classification performance with LiT on a wide range of datasets, locking the image tower is a drastic measure that introduces a severe dependency on the pretrained model, prohibiting the image tower from improving during contrastive training. We will later show that, if the embeddings in the frozen image tower are not suited to a particular downstream tasks, LiT underperforms compared to approaches that train the image tower on the varied contrastive learning dataset, see, for example, §4.2 and §4.4. The 3T approach seeks to address these concerns.

## 3 Three Towers: Flexible Contrastive Learning with Pretrained Models

With Three Towers (3T), we propose a simple and flexible approach to incorporate knowledge from pretrained models into contrastive learning. Instead of directly using the pretrained model locked as the main image tower, we instead add a *third* tower, $h$, which contains the fixed pretrained embeddings. The main image and text towers are trained from scratch, and we transfer representations from the third tower to the main towers with additional contrastive losses. In this setup, the main image tower benefits from *both* pretraining knowledge and contrastive learning.

More formally, in addition to the standard image and text towers, $f_\theta$ and $g_\phi$, cf. §2, we now have access to fixed pretrained image embeddings $p : \mathcal{I} \to \mathbb{R}^P$. Because $P$ can be different from the target dimension $D$, we define the third tower as $h(I) = \texttt{linear}(p(I))$, where $\texttt{linear} : \mathbb{R}^P \to \mathbb{R}^D$ projects

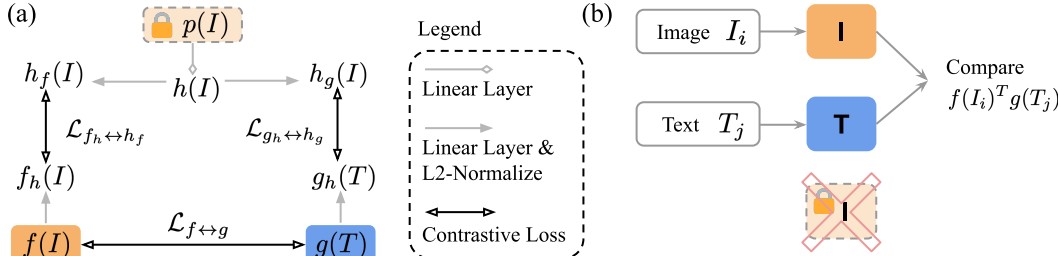

Figure 2: Details of the 3T approach. (a) Linear adaptor heads (gray) align the representations between the main towers and the third tower. (b) For downstream tasks, 3T is used in the same way as CLIP/ALIGN and LiT. We discard the third tower, using only the main towers, $f(I)$ and $g(T)$.

embeddings to the desired dimensionality. In principle, the 3T architecture is also compatible with pretrained text models. However, like Zhai et al. [85], we do not observe benefits from using pretrained text models, cf. §A.3, and so our exposition and evaluation focuses on pretrained image classifiers.

When computing loss terms involving the third tower, we make use of learned linear projection heads. These heads afford the model a degree of flexibility when aligning representations between towers. First, we define $\text{NL}(\cdot) = \text{norm}(\text{linear}(\cdot))$, where $\text{norm}(x) = x/\|x\|_2$ normalizes with respect to L2 norm and, overloading notation, $\text{linear} : \mathbb{R}^D \to \mathbb{R}^D$ now preserves dimensionality. We adapt the main image and text towers as $f_h(I) = \text{NL}(f(I))$ and $g_h(T) = \text{NL}(g(T))$. We project the third tower embeddings $h$ to $h_f(I) = \text{NL}(h(I))$ and $h_g(I) = \text{NL}(h(I))$ for computation of the loss with the image and text towers respectively. The linear layers introduced for $f_h$, $g_h$, $h_f$, and $h_g$ are independently learned from scratch. Per input batch, the 3T approach then optimizes the following loss objective:

$$\mathcal{L}_{3\text{T}} = \frac{1}{3} \cdot (\mathcal{L}_{f \leftrightarrow g} + \mathcal{L}_{f_h \leftrightarrow h_f} + \mathcal{L}_{g_h \leftrightarrow h_g}). \tag{4}$$

Here, $\mathcal{L}_{f \leftrightarrow g}$ is the original contrastive loss, cf. Eq. (1), and $\mathcal{L}_{f_h \leftrightarrow h_f}$ and $\mathcal{L}_{g_h \leftrightarrow h_g}$ are additional contrastive losses between the image/text tower and the third tower projections. All loss terms share a global temperature $\tau$. We train both towers, $f$ and $g$, and all linear layers from scratch by optimizing Eq. (4) over input batches. Figure 2 (a) visualizes the adaptor heads and loss computation for 3T.

After training, the third tower is discarded and we use only the main two towers, cf. Fig. 2 (b). Therefore, the inference cost of 3T is equal to the baseline methods. For training, the additional cost over the from-scratch CLIP/ALIGN baseline is negligible, as frozen embeddings from the third tower can be pre-computed and then stored with the dataset, as also done in Zhai et al. [83].

**Intuitions for the 3T Architecture.** Intuitively, the additional losses align the representations of the main towers to the pretrained embeddings in the third tower. In fact, Tian et al. [71] show that contrastive losses can be seen as *distillation* objectives that align representations between a teacher and a student model. They demonstrate that contrastive losses are highly effective for representation transfer, outperforming alternative methods of distillation. Thus, the additional terms, $\mathcal{L}_{f_h \leftrightarrow h_f}$ and $\mathcal{L}_{g_h \leftrightarrow h_g}$, transfer representations from the pretrained model to the unlocked main towers, albeit without the usual capacity bottleneck between the student and teacher models. Of course, for 3T, we also need to consider the original objective $\mathcal{L}_{f \leftrightarrow g}$ between the unlocked text and image towers. In sum, the unlocked towers benefit both from the pretrained model and contrastive training.

**Design Choices.** In §4.6, we ablate various design choices, such as our use of equal weights among the terms of Eq. (4), the contrastive loss for representation transfer, the shared global temperature $\tau$, the linear layers for $h$ in the third tower, as well as our design of the adaptor heads.

**Discussion.** The 3T approach includes pretrained knowledge without suffering from the inflexibility of directly using the pretrained model as the main image tower. For example, unlike LiT, 3T allows for architectural differences between the unlocked image tower and pretrained model. Further, it seems plausible that 3T should generally be more robust than LiT: as the image tower learns from the highly-diverse contrastive learning datasets, the chances of encountering 'blindspots' in downstream applications, e.g. due to labels or examples not included in the pretraining dataset, should be lower. In a similar vein, LiT is most appropriate for pretrained models *so capable* that they need not adapt during contrastive training. For example, few-shot classification performance, which uses only the image tower, by design cannot improve at all during contrastive training with LiT. On the other hand, with 3T

we may be able to successfully incorporate knowledge from weaker models, too. Lastly, finetuning—instead of locking—the main image tower from a pretrained classifier often has at most marginal positive effects over training from scratch after a high number of training steps, cf. §2. In contrast, the additional terms in Eq. (4) consistently ensure the main towers align with the pretrained model.

# 4 Experiments

In this section, we compare 3T to LiT and to a standard CLIP/ALIGN baseline trained from scratch, which we will refer to as the 'baseline' for simplicity. Our experimental setup largely follows Zhai et al. [85]: for all methods, we use Vision Transformers (ViT) [17] for the image and text towers, replacing visual patching with SentencePiece encoding [41] for text inputs, further sharing optimization and implementation details with [85]. We rely on the recently proposed WebLI dataset [11], a large-scale dataset of 10B image-caption pairs (Unfiltered WebLI). We also explore two higher-quality subsets derived from WebLI: Text-Filtered WebLI, which uses text-based filters following Jia et al. [34], and Pair-Filtered WebLI (see §A.7), which retains about half of the examples with the highest image-text pair similarity. For image tower pretraining, we consider both proprietary JFT-3B [84] and the publicly available IN-21k checkpoints of Dosovitskiy et al. [17]. For IN-21k experiments, our largest model scale is L, with a $16 \times 16$ patch size for ViT, and for JFT pretraining we go up to g scale at a patch size of $14 \times 14$. Unless otherwise stated, we train for 5B examples seen at a batch size of $14\,336$.

## 4.1 Retrieval

We study zero-shot retrieval performance on COCO [10] and Flickr [56]. Table 1 shows results for g scale models trained on Text-Filtered WebLI, with JFT pretraining for 3T and LiT. In Table 2, we report performance of L scale models trained on Unfiltered WebLI for JFT and IN-21k pretraining. We give results for additional WebLI splits for IN-21k and JFT pretraining in Tables A.4 and A.6.

Table 2: 3T outperforms LiT and the baseline for retrieval (top-1 recall $\uparrow$, L scale models, Unfiltered WebLI, see §4.1).

| Pretraining Method | – Basel. | IN-21k LiT | 3T | JFT LiT | 3T |
|---|---|---|---|---|---|
| Flickr* img2txt | 75.6 | 71.7 | 80.0 | 78.7 | 80.0 |
| Flickr* txt2img | 57.1 | 49.3 | 60.9 | 58.8 | 61.4 |
| COCO img2txt | 51.0 | 46.1 | 54.4 | 52.7 | 54.4 |
| COCO txt2img | 34.2 | 27.8 | 37.7 | 36.7 | 37.9 |

*3T improves on LiT and the baseline for retrieval tasks* across scales, datasets, and for both JFT and IN-21k pretraining. A rare exception to this are the Unfiltered WebLI results in Table A.6, where 3T beats LiT for retrieval on average and for txt2img, but not for img2txt. In general however, LiT underperforms for retrieval and regularly does not outperform the baseline: at L scale, LiT shows a strong dependence on the pretrained model and can only outperform the baseline with JFT pretraining. In contrast, 3T obtains similar improvements over the baseline for both pretraining datasets. We will continue to see this pattern in our experiments: 3T consistently improves over the baseline, while LiT results can vary wildly and depend strongly on the pretraining dataset. We discuss our retrieval results in the context of SOTA performance in §A.7: the SOTA method CoCa [81] achieves better results but uses about 4 times more compute; increasing the compute budget for 3T would likely reduce the gap.

Intuitively, while the fixed classifier embeddings in LiT can categorize inputs into tens of thousands of labels, they may not be fine-grained enough for retrieval applications. On the other hand, the contrastive training of the baseline is closely related to the retrieval task but misses out on knowledge from pretrained models. Only 3T is able to combine benefits from both for improved performance.

## 4.2 Few-Shot and Zero-Shot Image Classification

Next, we compare the approaches on few- and zero-shot image classification. See §D for citations of all the datasets that we use. For zero-shot classification, we follow the procedure described in §2. For few-shot tasks, we report 10-shot accuracy, more specifically, the accuracy of a linear classifier trained on top of fixed image representations, averaging over 3 seeds for the 10 random examples per class. As few-shot performance depends only on the image embeddings, LiT's few-shot accuracy is precisely the same as that of the pretrained model. Despite this, Zhai et al. [85] show that LiT outperforms the unlocked baseline on few-shot IN-1k and CIFAR-100 evaluations.

---

*For historical reasons, *Flickr** results do not use the 'Karpathy' split [36]. However, they are available for all runs, and they are directly comparable. Our g scale runs do have Karpathy split results, denoted *Flickr* (no star).

For IN-21k pretraining, Table 3 reports the performance of L scale models trained on Unfiltered WebLI, and we give results on additional datasets in Table A.4. Here, 3T outperforms both LiT and the baseline. For JFT pretraining, Table A.5 gives results at L scale on Unfiltered WebLI and Table A.6 presents results at g scale across WebLI splits. In all JFT settings, LiT performs best on average for image classification tasks, despite few-shot performance being fixed for LiT. However, for both JFT and IN-21k pretraining, 3T improves over the baseline for almost all tasks. This is different for LiT, where performance heavily depends on the pretraining data.

**Risk of Locking.** There is a risk associated with LiT, both in a positive and negative sense. Using fixed classifier representations can result in excellent performance if the downstream task distribution and pretraining dataset are well-aligned: for example, IN-21k contains hundreds of labels of bird species, and IN-21k-LiT performs well on the Birds task, outperforming 3T by $18\,\%$p. However, the IN-21k label set does not contain a single car brand, and thus, IN-21k-LiT does not perform well on Stanford Cars, underperforming relative to the *baseline* and 3T by almost $40\,\%$p. ObjectNet, IN-A, and IN-R were created to be challenging for ImageNet models, and so IN-21k-LiT performs worse than the baseline and 3T here, too. For example, IN-R contains artistic renditions of objects that are challenging for IN-21k-based models as they have mostly been trained on realistic images. IN-21k-LiT also struggles with more specialized tasks, performing $29\,\%$p worse than the baseline on the remote sensing RESISC dataset.

The above results support our hypothesis that image embeddings trained on the highly diverse contrastive learning dataset will be more broadly applicable. Strikingly, even when using the same IN-21k model, 3T almost always improves over the baseline and never suffers the same failures as LiT. However, it seems that JFT-pretrained models can fix many of LiT's shortcomings for image classification. JFT-LiT performs remarkably well and almost never underperforms significantly compared to the baseline. A deviation from this pattern are the results for Eurosat and RESISC at g scale for JFT pretraining in Table A.6, where, e.g. on the Text-Filtered WebLI split, LiT lacks behind 3T by $12\,\%$p and $8\,\%$p.

Table 3: For IN-21k pretraining, 3T has the best average classification accuracy ($\uparrow$) (L scale models, Unfiltered WebLI).

| | Method | Basel. | LiT | 3T |
|---|---|---|---|---|
| Few-Shot Classification | IN-1k | 62.8 | 79.0 | 68.0 |
| | CIFAR-100 | 70.4 | 83.6 | 72.5 |
| | Caltech | 91.0 | 88.4 | 92.3 |
| | Pets | 85.9 | 89.2 | 86.5 |
| | DTD | 70.3 | 69.2 | 73.3 |
| | UC Merced | 91.8 | 92.8 | 94.0 |
| | Cars | 81.5 | 41.9 | 84.9 |
| | Col-Hist | 71.7 | 86.4 | 76.6 |
| | Birds | 53.4 | 83.4 | 65.0 |
| Zero-Shot Classification | IN-1k | 69.5 | 76.0 | 71.7 |
| | CIFAR-100 | 73.5 | 82.9 | 73.4 |
| | Caltech | 81.9 | 82.4 | 84.1 |
| | Pets | 84.2 | 87.1 | 87.0 |
| | DTD | 58.6 | 51.8 | 60.3 |
| | IN-C | 49.6 | 62.0 | 51.8 |
| | IN-A | 53.0 | 45.6 | 54.3 |
| | IN-R | 85.8 | 66.1 | 88.1 |
| | IN-v2 | 62.2 | 67.2 | 64.9 |
| | ObjectNet | 56.2 | 41.9 | 58.3 |
| | EuroSat | 32.7 | 27.6 | 42.8 |
| | Flowers | 62.0 | 72.6 | 65.7 |
| | RESISC | 58.0 | 29.0 | 57.9 |
| | Sun397 | 67.6 | 65.4 | 68.7 |
| **Average** | | 68.4 | 68.3 | 71.4 |

## 4.3 Scaling Model Sizes and Training Duration

Since the publication of LiT, the scale of contrastive learning datasets, both public and proprietary, has increased, for example with the release of LAION-5B [64] or WebLI-10B [11]; we use the latter here. Given their cheap collection costs, it seems likely that this growth will continue to outpace that of more expensive classification datasets. However, locking the image tower ignores any potential benefits from the increased contrastive learning data for the image tower. Additionally, larger datasets often lead to increases in model scales to fully make use of the additional information [84]; unlike LiT, 3T can increase the scale of the main image tower independently of the pretrained model, cf. §4.4.

Here, we separately study the effects that model scale and dataset size have on 3T, LiT, and the baseline. First, we vary the scale of all involved towers, including the pretrained model. We study both IN-21k and JFT pretraining, always train contrastively for 5B examples seen, and the S, B, L, and g scales correspond to S/32, B/32, L/16, and g/14 for the ViT models. We compute averages separately across retrieval, few-shot, and zero-shot classification tasks, where the tasks are those from Tables 2 and 3.

In Fig. 3, we observe that 3T's lead over LiT and the baseline in retrieval performance holds across scales and pretraining datasets. Further, across all tasks and scales, 3T maintains a consistent performance gain over the baseline. Also across tasks and pretraining datasets, we observe that

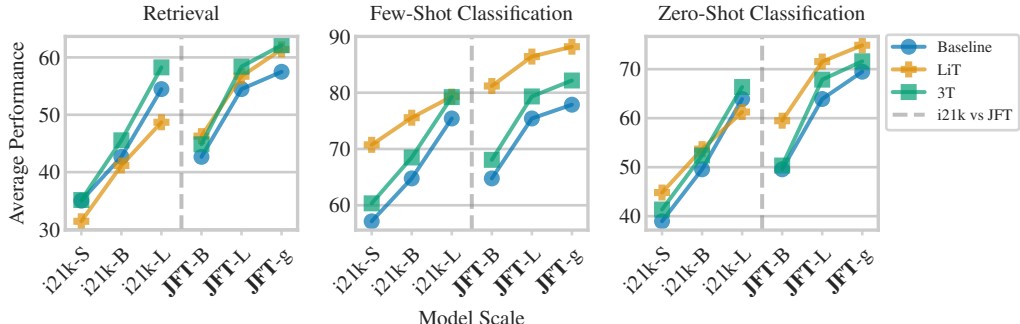

Figure 3: We increase the model scale for 3T, LiT, and the baseline and report average retrieval, few- and zero-shot classification performance for both IN-21k and JFT pretraining. 3T and the baseline benefit more from increases in scale than LiT (their curves are steeper), with 3T performing better than the baseline. The baseline does not use a pretrained model and is displayed twice (at scale B and L).

scaling is more beneficial for 3T than for LiT: as we increase the scale, 3T's performance increases more than that of LiT. This means that 3T either extends its lead over, overtakes outright, or reduces its gap to LiT as we increase scale. At L scale with IN-21k pretraining, 3T gives the best average performance of all methods across tasks. For JFT pretraining, LiT maintains an edge for classification performance, although the scaling behavior suggests this gap may fully collapse at larger scales. We observe similar trends when scaling the number of examples seen during training, cf. Fig. A.4.

## 4.4 Pretraining Robustness

Next, we study what happens when 3T and LiT are used with pretrained models that do not conform to expectations. We consider two setups: one that we call 'mismatched' and one that considers pretraining on the Places365 [87] dataset. In Table 4, we display zero-shot accuracies on Pets, IN-1k, and CIFAR-100—tasks for which LiT usually performs best—as well as the average performance over the full set of tasks, see Table A.7 for individual results.

Table 4:  3T is more robust to the pretraining setup than LiT. Zero-shot classification accuracies (↑), full details in main text.

| Setup | Mismatched | | | Places365 | | |
|---|---|---|---|---|---|---|
| Method | Basel. | LiT | 3T | Basel. | LiT | 3T |
| IN-1k | 69.5 | 69.5 | 71.5 | 45.6 | 24.5 | 47.4 |
| CIFAR-100 | 73.5 | 78.6 | 75.6 | 48.3 | 27.4 | 52.4 |
| Pets | 84.2 | 84.7 | 87.4 | 61.5 | 30.3 | 60.2 |
| ... | | ... | | | ... | |
| **Full Average** | 66.4 | 61.7 | 69.8 | 47.5 | 29.4 | 49.3 |

**Mismatched Setup.** So far, we have always matched the scale of the pretrained model to the scale of the models trained contrastively. For the mismatched setup, we now break this symmetry: we use an IN-21k-pretrained B/32 scale image model (3T and LiT) with an L scale text tower (all approaches) and an L/16 unlocked image tower (3T and baseline). This setup is relevant when pretrained models are not available at the desired scale: for example, given ever larger contrastive learning datasets one may want to train larger image models than are available from supervised training, cf. §4.3. Of course, increasing the image tower scale also comes at increased compute costs for 3T and the baseline. We observe that LiT suffers from this mismatched setup much more than 3T, which is not restricted by the smaller pretrained model and now achieves higher accuracy than LiT on Pets and IN-1k.

**Places365.** Zhai et al. [85] demonstrate that LiT performs badly when used with models pretrained on Places365. In Table 4, we reproduce this result and observe LiT performing much worse than the baseline. (We here train B scale models for 900M examples seen, and based on our discussion in §4.3, would expect LiT to perform even worse, in comparison to the baseline and 3T, when training longer or with larger scale models.) The embeddings obtained from Places365 pretraining do not allow LiT to perform well on our set of downstream tasks. 3T behaves much more robustly and does not suffer from any performance collapse because it incorporates both the pretrained model and contrastive data when training the image tower. Notably, 3T manages to improve average performance over the baseline even for Places365 pretraining. We further suspect the linear projection heads afford 3T some flexibility in aligning to the pretrained model without restricting the generality of the embeddings learned in the main two towers.

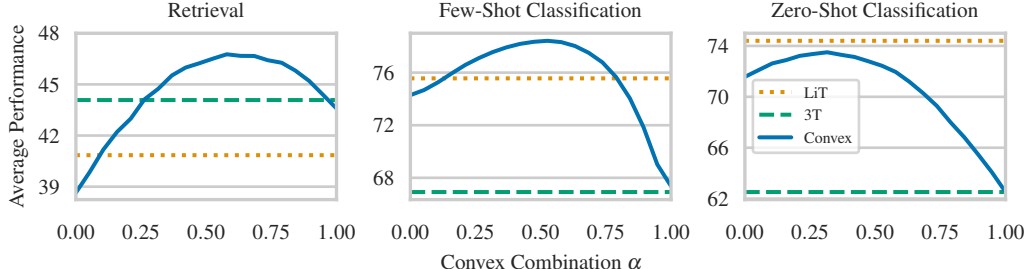

Figure 4: Convex combination of the image models in 3T: $\alpha \cdot h(I) + (1-\alpha) \cdot f(I)$. By varying $\alpha$, we can generally interpolate between 3T and LiT performance. Interestingly, for a broad range of weights, the retrieval and few-shot classification performance of the combination outperforms 3T and LiT.

## 4.5 Benefits From Using 3T With All Three Towers at Test Time

We usually discard the pretrained model when applying 3T to downstream tasks, cf. Fig. 2 (b). In this section, we instead explore if we can find benefits from using the locked third tower at test time, similar to LiT. More specifically, we study the convex combination of the main image tower and locked pretrained model in the third tower, $\alpha \cdot h(I) + (1-\alpha) \cdot f(I)$, to see if we can interpolate between 3T- and LiT-like prediction at $\alpha = 0$ and $\alpha = 1$ respectively. We train 3T without linear projection heads to make embeddings from all towers compatible. Additional details of this setup can be found in §A.10.

Fig. 4 shows we can generally interpolate between LiT- and 3T-like performance as we vary $\alpha$ from 0 to 1. Note that we do not always recover LiT or 3T performance at $\alpha \in \{0, 1\}$ as explained in §A.10. Interestingly, for retrieval and few-shot classification tasks, the convex combination yields better performance than either of the underlying methods for a relatively broad region around $\alpha \approx 0.5$. We believe that further study of the convex combination could be exciting future work: the method is entirely post-hoc and no additional training costs are incurred, although inference costs do increase.

## 4.6 Ablations

Next, we provide ablations for some of the design decisions of 3T as well as insights into LiT training. We perform the ablation study at B scale with patch size 32, training for 900M examples seen, and use JFT-pretrained models. In Table 5, we report the average difference in performance to our default runs across all tasks, together with two standard errors computed over the downstream tasks as an indication of statistical significance. We refer to §A.11 for full details and results.

**3T Ablations.** 'Rerun': To study per-run variance, we perform a rerun of the base 3T model, obtaining an average performance difference of $-0.22\,\%\mathrm{p}$ across tasks. 'No $\mathcal{L}_{\cdot \leftrightarrow \cdot}$': When leaving out either of the three loss terms, average performance suffers significantly. 'Head Variants': We try a selection of different projection head variants, see §A.11. None give significantly better performance than our default setup. 'MLP Embedding': Replacing the linear projection $h$ in the third tower with an MLP does not improve performance. 'More Temperatures': Using three learned temperatures, one per 3T loss term, instead of a global temperature as in Eq. (4), does not improve results. 'Loss weights': Replacing

Table 5: Ablation study, see text for details.

| | Difference to 3T |
|---|---|
| Rerun | $-0.22 \pm 0.25$ |
| No $\mathcal{L}_{f \leftrightarrow g}$ Loss | $-26.63 \pm 10.61$ |
| No $\mathcal{L}_{f_h \leftrightarrow h_f}$ Loss | $-1.19 \pm 0.75$ |
| No $\mathcal{L}_{g_h \leftrightarrow h_g}$ Loss | $-2.77 \pm 0.91$ |
| Head Variants | $0.09 \pm 0.35$ |
| MLP Embedding | $-0.08 \pm 0.35$ |
| More Temperatures | $-0.26 \pm 0.48$ |
| Loss Weights | $0.17 \pm 0.53$ |
| L2 Transfer | $-3.80 \pm 1.13$ |
| 3T Finetuning | $1.85 \pm 1.27$ |

| | Difference to LiT |
|---|---|
| Rerun | $-0.10 \pm 0.22$ |
| LiT Finetune | $-14.99 \pm 6.09$ |
| FlexiLiT1 | $-4.63 \pm 1.36$ |
| FlexiLiT2 | $-5.04 \pm 1.54$ |

the loss with a weighted objective, $\frac{1}{3} \cdot (\mathcal{L}_{f \leftrightarrow g} + w \cdot (\mathcal{L}_{f_h \leftrightarrow h_f} + \mathcal{L}_{g_h \leftrightarrow h_g}))$, does not improve performance significantly across a variety of choices for $w$. 'L2 Transfer': Using squared losses for the representation transfer objectives $\mathcal{L}_{f_h \leftrightarrow h_f}$ and $\mathcal{L}_{g_h \leftrightarrow h_g}$, cf. [60], results in significantly worse performance, even when optimizing the weight of the transfer terms. '3T Finetuning': Initializing the main tower in 3T with the same JFT-pretrained model as the third tower increases performance significantly; however, we find this effect becomes negligible for larger scale experiments, cf. §A.11.

**LiT Ablations.** 'Rerun': We observe similar between-run variance for LiT. 'LiT Finetune': We confirm the result of Zhai et al. [85] that finetuning from a pretrained model results in worse performance than locking. 'FlexiLiT 1/2': We investigate simple ways of modifying LiT such that it can adjust the image tower during contrastive learning, see §A.11, but find these are not successful.

**Additional Experiments.** In §A.1, we study the optimization behavior of 3T, finding evidence for beneficial knowledge transfer from the pretrained model. In §A.2, we study the calibration of all methods for zero-shot classification, as well as their performance for out-of-distribution (OOD) detection: 3T is generally better calibrated than LiT, and for OOD tasks, we find trends similar to §4.3, with all methods generally performing well. In §A.3, we confirm the results of Zhai et al. [85] that there are no benefits from using pretrained *language* models. In §A.4, a detailed investigation of predictions suggests that 3T performs well because it combines knowledge from contrastive learning and the pretrained model. Lastly, §A.5 shows 3T continues to perform well with other pretrained image models.

## 5   Related Work

CLIP [58] and ALIGN [34] are examples of vision-language models that have received significant attention, e.g. for their impressive ImageNet zero-shot results. Concurrently with LiT [85], BASIC [55] investigates locking and finetuning from JFT-pretrained models. Previously, [46, 57, 74, 67] have explored representation learning from images with natural language descriptions before the deep learning era. Subsequently, [21, 36, 35, 19, 63, 8] explore image-text understanding with CNNs or Transformers. In this context, Li et al. [42] introduced the idea of zero-shot transfer to novel classification tasks. The loss objective, Eq. (1), was proposed by Sohn [66] for image representation learning, and also appears in [77, 51, 9]. Zhang et al. [86] then used the objective to align images and captions, although their setting used medical data. Lots of work has built on CLIP and ALIGN. For example, [82, 79] have augmented the objective to optionally allow for labels, [89, 88] proposed methods for improving zero-shot prompts, [2, 43] applied CLIP to video, [54, 49, 62, 33] used CLIP embeddings to improve generative modelling, and [47] study different ways of incorporating image-only self-supervision objectives into CLIP-style contrastive learning. Relatedly, vast amounts of work have explored self-supervised or contrastive representation learning of images only, e.g. [16, 25, 24, 6]. Transfer learning [52] applies embeddings from large-scale (weakly) labelled datasets to downstream task [68, 44, 38].

## 6   Limitations, Impact, and Conclusions

**Limitations.** While 3T consistently improves over LiT for retrieval tasks, for classification, 3T outperforms LiT with ImageNet-21k-pretrained models only at large scales, and may require even larger scales for JFT pretraining. Further, while inference costs are equal for all methods, 3T incurs additional training costs compared to LiT. We have compared methods at matching inference cost for simplicity because there are many ways to account for the cost of pretraining and embedding computation.

**Impact.** We believe that locking is a suboptimal way to incorporate pretrained image models, and we have demonstrated clear benefits from exposing the image tower to both the contrastive learning dataset and the pretrained model, particularly as scale increases. 3T is a simple and effective method to incorporate pretrained models into contrastive learning and should be considered by future research and applications whenever strong pretrained models are available. For future work that seeks to improve 3T, we consider it important to understand the differences between embeddings from 3T, LiT, and the baseline. If we can obtain insights into why they excel at different tasks, we can perhaps (learn to) combine them for further performance improvements. Our convex combination experiments are a starting point; it would be interesting to continue this direction, possibly looking at combinations in parameter space [75, 76]. Lastly, future work could explore 3T for distillation of large pretrained models into smaller models, extend 3T to multiple pretrained models, potentially from diverse modalities, or explore the benefits of 3T-like ideas for other approaches such as CoCa [81].

**Conclusions.** We have introduced the Three Tower (3T) method, a straight-forward and effective approach for incorporating pretrained image models into the contrastive learning of vision-language models. Unlike the previously proposed LiT, which directly uses a frozen pretrained model, 3T allows the image tower to benefit from both contrastive training and embeddings from the pretrained model. Empirically, 3T outperforms both LiT and the CLIP/ALIGN baseline for retrieval tasks. In contrast to LiT, 3T consistently improves over the baseline across all tasks. Further, for ImageNet-21k-pretrained models, 3T also outperforms LiT for few- and zero-shot classification. We believe that the robustness and simplicity of 3T makes it attractive to practitioners and an exciting object of further research.

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

# Appendix

# Three Towers: Flexible Contrastive Learning
# with Pretrained Image Models

## A  Additional Experiments & Results

### A.1  Training Dynamics

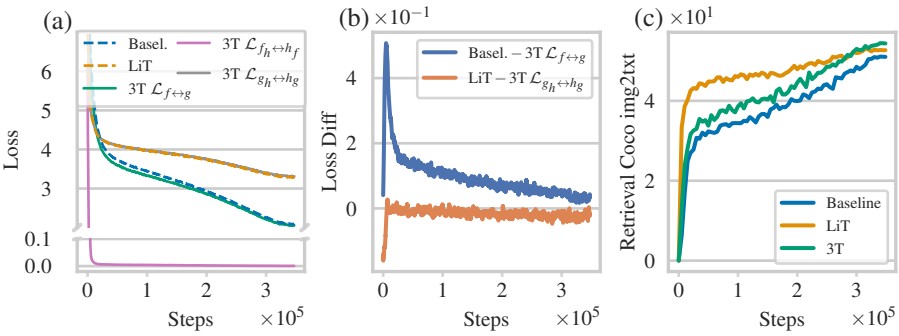

Figure A.1: Training dynamics: (a) The transfer losses, $\mathcal{L}_{f_h \leftrightarrow h_f}$ and $\mathcal{L}_{g_h \leftrightarrow h_g}$, improve the image-text loss, $\mathcal{L}_{f \leftrightarrow g}$, in 3T relative to the baseline. (b) Difference between matching loss terms for 3T, LiT, and the baseline. 3T obtains better image-to-text loss than the baseline and similar locked-image-to-text loss as LiT. (c) While the loss advantage of 3T over the baseline shrinks during training, this does not happen for downstream applications; we display image-to-text retrieval on COCO as an example. Moving averages applied to (a-b) for legibility.

In Fig. A.1, we compare the 3T training losses to LiT and the from-scratch baseline, using the familiar L scale setup with JFT pretraining. For 3T, we display all loss terms separately: the image-text loss $\mathcal{L}_{f \leftrightarrow g}$, the image-to-third-tower loss $\mathcal{L}_{f_h \leftrightarrow h_f}$, and the text-to-third-tower loss $\mathcal{L}_{g_h \leftrightarrow h_g}$, cf. Eq. (4) and Fig. 2. For LiT and the baseline, there is only the image-to-text loss as per Eq. (1). As we train for less than one epoch, we do not observe any overfitting, in the sense that contrastive losses are identical on the training and validation set.

The image-to-third-tower loss, $\mathcal{L}_{f_h \leftrightarrow h_f}$, quickly reduces to near zero, indicating successful knowledge transfer of the pretrained model into the image tower for 3T. Further, $\mathcal{L}_{f \leftrightarrow g}$ behaves similar to the baseline loss; this makes sense because both objectives compute a loss between an unlocked image tower and a text tower. Lastly, $\mathcal{L}_{g_h \leftrightarrow h_g}$ closely follows LiT's loss; this also makes sense because both are losses between a locked pretrained image model and a text tower trained from scratch.

In Fig. A.1 (b), we compute the difference of the baseline (LiT) loss and matching 3T $\mathcal{L}_{f_h \leftrightarrow h_f}$ ($\mathcal{L}_{g_h \leftrightarrow h_g}$) loss, and observe that 3T generally achieves lower (similar) values for the same objective. This suggests a mutually beneficial effect for the individual loss terms of the 3T objective. By aligning the main image and text towers to the pretrained model, 3T obtains improved alignment between the main towers themselves. For the training loss, this effect is large early in training and then decreases. However, for downstream task application, we find that the gap between 3T and the baseline persists; we display retrieval on COCO as an example in Fig. A.1 (c).

### A.2  Robustness Metrics

In this section, we study 3T, LiT, and the from-scratch baseline from a robustness perspective, evaluating on a subset of the tasks considered by Tran et al. [72]. Following §4.3, we evaluate all methods across multiple model scales and for both JFT and IN-21k pretraining. We use the full Unfiltered WebLI for all results here. We apply models in zero-shot fashion to these datasets, following the same protocol as for the main zero-shot classification experiments. We continue to use the global temperature $\tau$, cf. §3, learned during training to temper the probabilistic zero-shot predictions.

### A.2.1 Probabilistic Prediction and Calibration on CIFAR and ImageNet Variants

In Fig. A.2, we report accuracy, negative log likelihood (NLL), Brier score [22], and expected calibration error (ECE) [48, 22] for 3T, LiT, and the baseline across scales for the following datasets: CIFAR-10, CIFAR-10-C, ImageNet-1k (IN-1k), IN-A, IN-v2, IN-C, and IN-R.

**Accuracy.** Across all datasets, we find the familiar scaling behavior discussed in §4.2: 3T is consistently better than the baseline, 3T benefits more from increases in scale than LiT, LiT performs well with JFT pretraining but shows weaknesses when pretrained on IN-21k. Note that, for the ImageNet variants, we have previously reported the accuracies (if only at L scale) in §4.2. (Note further, that there might be small discrepancies, because we actually recompute all numbers from a different codebase for the robustness evaluations [15].) For CIFAR-10 [40] and CIFAR-10-C [28], which we have not previously discussed, we also find the familiar scaling behavior. The absolute reduction of performance between CIFAR-10 and CIFAR-10-C is similar across methods, indicating that no approach is significantly more robust to shifts. We observe the same comparing IN-1k to IN-C.

**Probabilistic Prediction and Calibration.** NLL and Brier scores follow the general trend laid out by the accuracy results. Evidently, the *probabilistic* zero-shot predictions of the methods are all of similar high quality, cf., for example, Tran et al. [72], who investigate probabilistic few-shot predictions. This is confirmed by the ECE results: across tasks, ECE values do not exceed $0.1$ at L scale. For 3T, calibration results are regularly better than for LiT, particularly if pretrained on IN-21k, and comparable to those of the baseline: 3T and the baseline have lower calibration error than LiT on 6 out of 7 tasks at L scale with IN-21k pretraining.

We find the low magnitude of the calibration errors surprising. It is striking that the softmax temperature learned during contrastive training would work so well across the various downstream task applications. After all, finding matches across a batch from the contrastive learning dataset and assigning images to labels are, at least superficially, quite distinct tasks. We stress again that no task adaptation of either the models, prompt templates, or softmax-temperatures was performed. We refer to Minderer et al. [45] for a general categorization of our calibration results and discussion in the context of deep learning models.

### A.2.2 Out-Of-Distribution Detection

We evaluate the performance of 3T, LiT, and the baseline for out-of-distribution (OOD) detection. We follow the common practice of thresholding the maximum softmax probabilities (MSP) of the models to obtain a binary classifier into in- and out-of-distribution [30, 20]. We report the following metrics: area under the precision-recall curve (AUC(PR)), the area under the receiver operating curve (AUROC), as well as the false positive rate at $95\%$ true positives (FPR95). Following Tran et al. [72], we study CIFAR-10 as in-distribution against CIFAR-100, DTD, Places365, and SVHN as out-of-distribution. We also report numbers for IN-1k (in-distribution) vs. Places365 (out-of-distribution).

Typically for OOD evaluations, the model is *trained* on the in-distribution data. Here, we apply methods in a zero-shot manner: we only condition the text tower on the label set of the particular in-distribution dataset. Our image and text towers are trained on the contrastive learning data (image tower trained on JFT/IN-21k for LiT) and *not* adapted to the in-distribution samples. Our contrastive learning methods 'learn' about the in-distribution data only through the label set, and they have to classify each incoming sample as 'in-distribution' or 'out-of-distribution' based solely on how well it aligns with the given set of labels. If a given sample does not match any of the in-distribution labels well, prediction confidence is low, and the sample is classified as OOD. This setup diverges from typical assumptions about OOD experiments and should be interpreted with care. For example, if there were label overlap between the in- and out-of-distribution data (e.g. as would be the case for SVHN vs. MNIST), it would be impossible for the model to classify between in-distribution and OOD without further assumptions. OOD for CLIP/ALIGN-style models has been studied in similar settings by Fort et al. [20], Esmaeilpour et al. [18].

We display results in Fig. A.3. Generally, OOD detection works well with the contrastively learned models, despite conditioning only on the label set: for example, the AUROC for CIFAR10 exceeds 0.95 for both 3T and LiT at L scale for both IN-21k and JFT pretraining. The different metrics, AUC(PR), AUROC, and FPR95, are generally consistent in their ranking across scales and methods. We again find the familiar pattern: 3T is consistently improving over the baseline, and 3T catches up

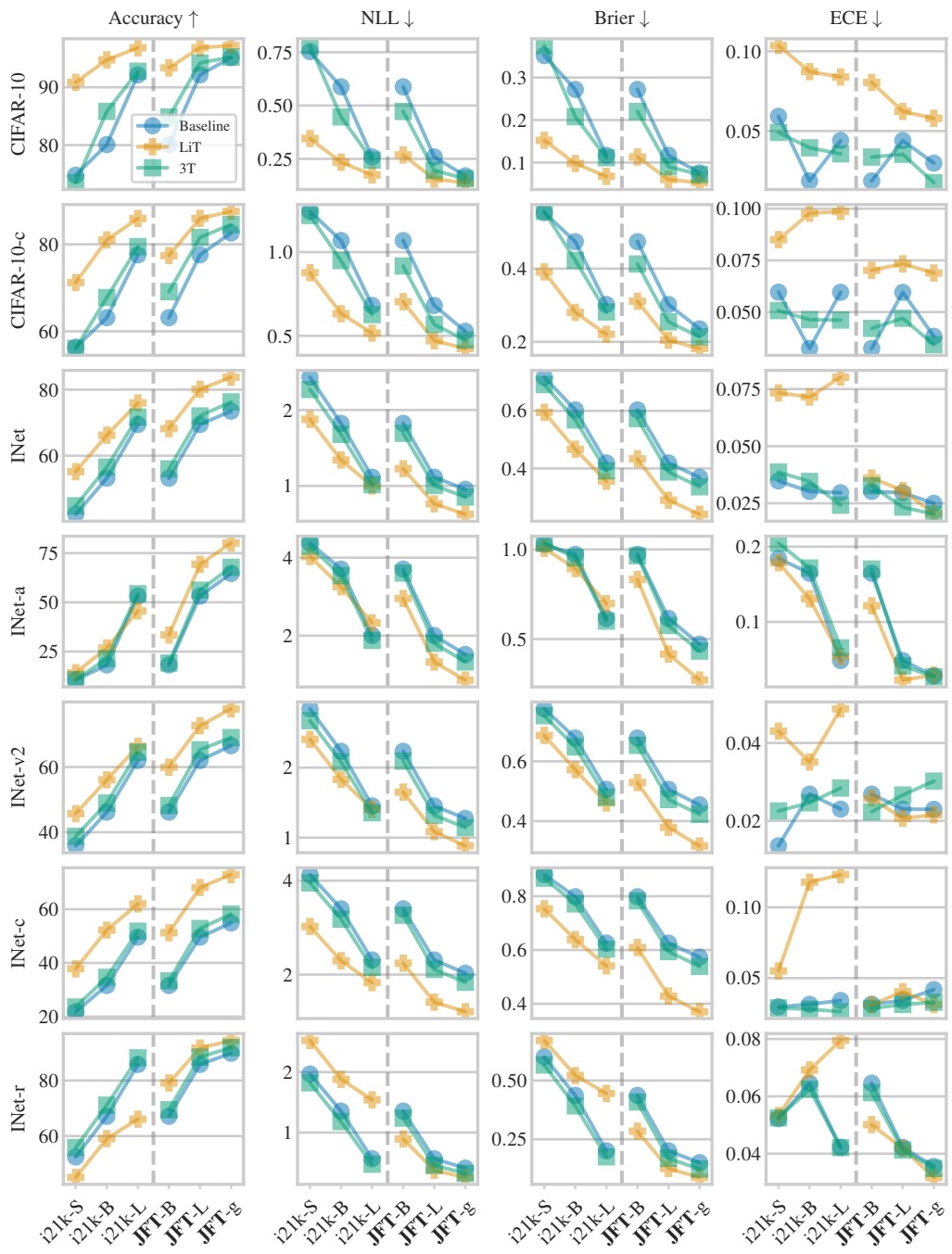

Figure A.2: Robustness evaluation: Accuracy, negative log likelihood (NLL), Brier score, and expected calibration error (ECE) for 3T, LiT, and the baseline for IN-21k and JFT pretraining across scales.

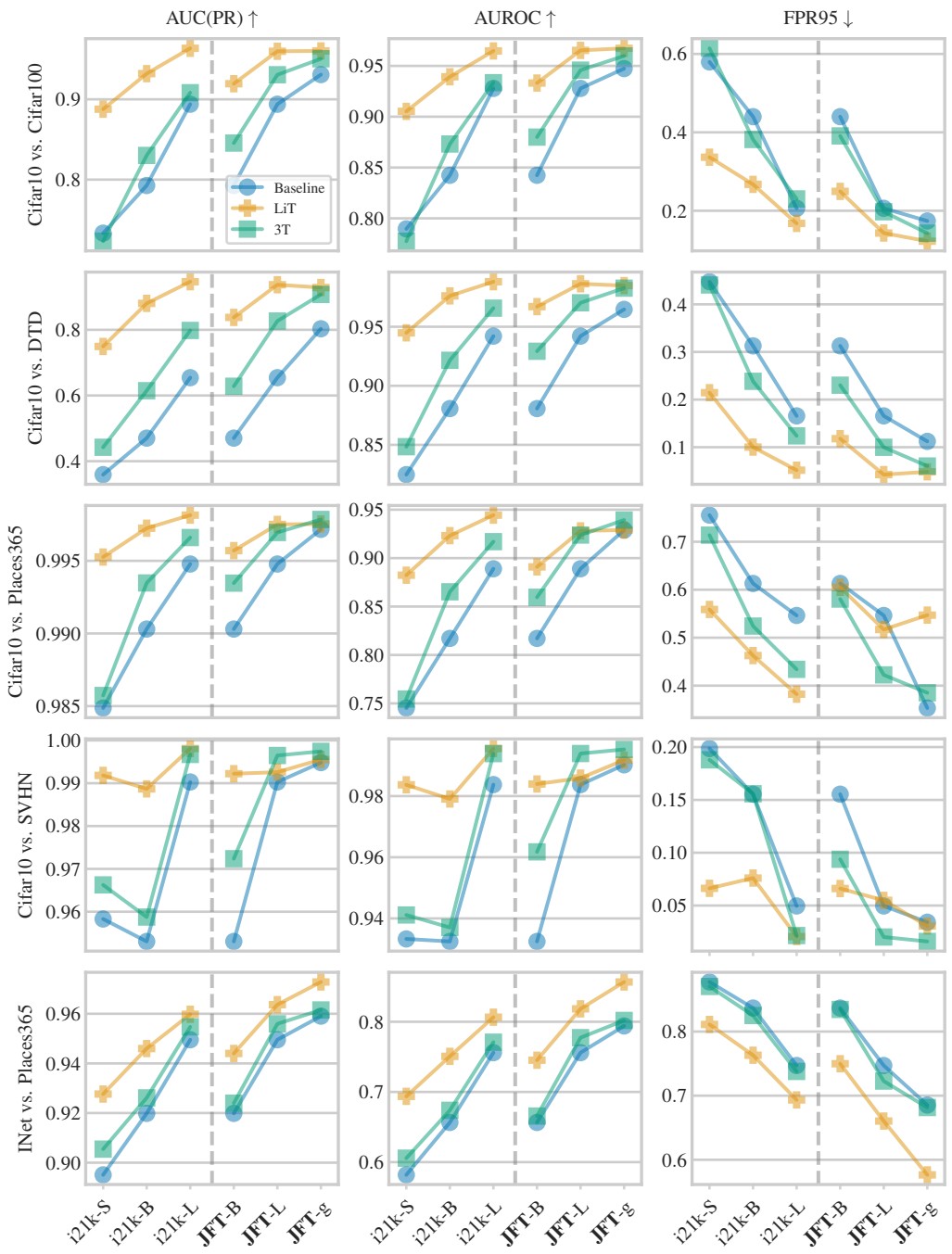

Figure A.3: Robustness evaluation: 3T, LiT, and the baseline for zero-shot out-of-distribution detection (OOD). Reported metrics are area under the precision-recall curve (AUC(PR)), the area under the receiver operating curve (AUROC), and the false positive rate at $95\%$ true positives (FPR95).

Table A.1: Using pretrained language models instead of image encoders degrades performance drastically for LiT, confirming the results of Zhai et al. [85]. While 3T does not suffer a drastic collapse in performance, we also do not observe gains from using the pretrained language model. B/32 ViT image tower, BERT models at BASE scale for LiT and 3T, B scale unlocked text tower for 3T and baseline, training for 900M examples seen at batch size 10 240.

| | | Unfiltered WebLI | | | Pair-Filtered WebLI | | |
| | | Basel. | LiT | 3T | Basel. | LiT | 3T |
|---|---|---|---|---|---|---|---|
| Retrieval | Flickr* img2txt | 55.2 | 4.4 | 50.6 | 65.7 | 10.6 | 64.4 |
| | Flickr* txt2img | 36.2 | 2.4 | 32.4 | 45.2 | 5.2 | 43.7 |
| | Coco img2txt | 34.5 | 2.2 | 29.9 | 42.8 | 5.0 | 38.4 |
| | Coco txt2img | 20.7 | 1.4 | 18.2 | 26.8 | 3.2 | 24.6 |
| Few-Shot Classification | Caltech | 87.4 | 80.3 | 87.0 | 88.7 | 88.0 | 90.0 |
| | UC Merced | 84.2 | 67.7 | 81.4 | 90.5 | 81.0 | 87.8 |
| | Cars | 56.8 | 38.0 | 56.8 | 76.9 | 67.1 | 76.8 |
| | DTD | 59.2 | 45.8 | 56.3 | 63.8 | 58.3 | 63.1 |
| | Col-Hist | 72.0 | 59.3 | 69.0 | 75.8 | 63.8 | 73.2 |
| | Birds | 32.1 | 16.9 | 29.5 | 48.5 | 35.1 | 46.7 |
| | Pets | 57.8 | 31.9 | 53.5 | 72.7 | 70.2 | 78.1 |
| | ImageNet | 37.8 | 24.2 | 35.2 | 46.5 | 41.6 | 47.9 |
| | CIFAR-100 | 50.2 | 33.8 | 45.2 | 57.5 | 43.7 | 53.3 |
| Zero-Shot Classification | Pets | 58.7 | 6.1 | 56.5 | 79.2 | 36.8 | 79.4 |
| | ImageNet | 45.8 | 11.3 | 41.8 | 58.0 | 26.9 | 56.9 |
| | CIFAR-100 | 52.2 | 19.1 | 44.0 | 57.6 | 37.6 | 54.7 |

to LiT as scale is increased. For OOD detection, LiT generally does better than 3T and the baseline, perhaps owing to the fact that our choice of CIFAR/IN-1k as in-distribution datasets is advantageous for LiT (similar to how LiT performs particularly well for these datasets for classification).

We find differences between JFT and IN-21k pretraining to be much smaller for the OOD detection task. In fact, in some cases, IN-21k pretraining outperforms JFT pretraining, for example with LiT for the CIFAR-10 vs. Places365 detection task. (This might again be due to the fact that IN-21k pretraining is sufficient for application to CIFAR-10, and only struggles to perform well for other, more varied datasets.) Further, we can observe a rare victory of 3T over JFT-LiT and the baseline at L and g scale in terms of FPR95 on CIFAR-10 vs. Places365 and CIFAR-10 vs. SVHN. Lastly, we see LiT has almost fixed performance at $\approx 0.98$ for the CIFAR-10 vs. SVHN task across scales, perhaps due to early task saturation.

### A.3 Pretrained Language Models

In Table A.1, we explore if there are benefits to using pretrained language models instead of pretrained image encoders. For LiT, we confirm the results of Zhai et al. [85] that performance suffers drastically when using a locked pretrained language encoder as the text tower (together with an unlocked image tower). In contrast, for 3T, we do not observe a drastic decrease in performance, once again showing that 3T is more robust to deficiencies in the pretrained model than LiT. However, we also do not observe gains from using a pretrained language model with 3T, justifying our choice of focusing on pretrained image encoders in the main text. Nevertheless, we think it is plausible that future work may yet find benefits of incorporating knowledge from pretrained language models in other settings or downstream tasks, e.g. tasks that require more complex language reasoning abilities.

Results here are for B/32 ViT image towers, we use BERT models at BASE scale as pretrained language models for LiT and 3T, a B scale unlocked text tower for 3T and the baseline, and we train for 900M examples seen at batch size 10 240.

Table A.2: Predictions between the baseline and LiT are different and 3T benefits from combining them. The first column gives the proportion of datapoints where the baseline but not LiT predicts correctly. The second column gives the proportion of datapoints where LiT but not the baseline predicts correctly. Columns three to five show the proportion of datapoints where 3T predicts differently than the baseline (3), LiT (4), or differently from both (5).

| | Base, Not LiT | LiT, Not Base | 3T $\neq$ Base | 3T $\neq$ LiT | 3T $\neq$ Base & 3T $\neq$ LiT |
|---|---|---|---|---|---|
| IN-1k | 6.9 | 13.3 | 22.8 | 26.5 | 14.5 |
| CIFAR-100 | 7.5 | 18.0 | 28.6 | 32.8 | 20.4 |
| Caltech | 3.9 | 4.3 | 11.5 | 16.1 | 8.4 |
| Pets | 7.2 | 7.1 | 10.9 | 14.4 | 5.5 |
| DTD | 14.3 | 7.6 | 27.2 | 40.3 | 19.5 |
| IN-A | 19.0 | 11.7 | 38.3 | 52.8 | 28.4 |
| IN-R | 23.5 | 3.8 | 13.3 | 34.2 | 10.0 |
| IN-v2 | 8.4 | 13.3 | 27.2 | 32.8 | 18.0 |
| ObjectNet | 20.7 | 6.4 | 34.1 | 52.9 | 26.7 |
| EuroSat | 19.2 | 14.2 | 62.3 | 75.1 | 49.4 |
| Flowers | 4.0 | 14.8 | 25.4 | 27.0 | 16.0 |
| RESISC | 33.2 | 4.1 | 31.9 | 62.6 | 25.7 |
| Sun397 | 11.8 | 9.6 | 21.9 | 31.2 | 14.2 |

## A.4 Investigating Prediction Differences

Table A.2 provides a detailed evaluation of how predictions differ between 3T, LiT, and the baseline on a per datapoint (and per task) level. The results in Table A.2 are for the zero-shot tasks of the IN-21k pretrained L scale model setup of Table 3. This evaluation gives meaningful insights into understanding how 3T improves predictions over LiT and the baseline. The contrastive learning baseline and the pretrained model have different strengths, and 3T can benefit from combining them.

The first two columns of Table A.2 show that there is predictive diversity between the baseline and LiT. The first column gives the proportion of datapoints where the baseline but not LiT predicts correctly. The second column gives the proportion of datapoints where LiT but not the baseline predicts correctly. We can see that, for both the baseline and LiT, there is a significant proportion of inputs, where only one of the models predicts correctly. Hence, the two models have different strengths and, equivalently, make different mistakes. A combination of the two approaches, such as 3T, can benefit from this if it learns to combine their predictions in the right way.

In columns three, four, and five, we illustrate that 3T predicts differently from LiT and the baseline. The columns show the proportion of datapoints where 3T predicts differently than the baseline (3), LiT (4), or differently from both (5). Clearly, 3T learns a novel predictive mechanism that is meaningfully different from LiT and the baseline. Whenever 3T outperforms both LiT and the Baseline (Caltech, DTD, IN-A, IN-R, ObjectNet, EuroSat, and Sun397 in Table 3) it must have learned to combine knowledge from the pretrained model and contrastive learning mechanism in a beneficial way.

## A.5 Additional Image Encoders

In Table A.3, we study 3T and LiT for additional pretrained image encoders: a ResNet-based BiT model trained on IN-21k [38] and a ViT-based self-supervised encoder trained with DINO on IN-1k as in [85]. 3T consistently outperforms the baseline and LiT for DINO- and BiT-based pretrained models for all retrieval scenarios. For few-shot classification, LiT performs best on average for BiT-based models but 3T performs best on average for our DINO experiments. For zero-shot classification, 3T performs best on average for both our DINO- and BiT-based experiments. In total, 3T performs best on average over all tasks for both experiment setups.

Here, we use a B/16 scale image tower and a B scale text encoder trained on Unfiltered WebLI. The pretrained image models are a BiT-R50x3 pretrained on IN-21k, and we follow Zhai et al. [85] for DINO pretraining on IN-1k. We also report a 'BiT-Baseline': for this we replace the ViT B/16 image tower of the standard baseline with a BiT-R50x3 trained from scratch.

Table A.3: Results for additional pretrained image encoders: a ResNet-based BiT trained on IN-21k and a self-supervised DINO encoder trained on IN-1k. 3T outperforms LiT and the baseline on average for retrieval and classification task, with the exception of few-shot classification for BiT, where LiT performs best. B/16 scale image tower, B scale text encoder, BiT-R50x3 pretrained on IN-21k, DINO pretrained on IN-1k as in Zhai et al. [85], Unfiltered WebLI.

| | | BiT Experiments | | | | DINO Experiments | | |
| | | Basel. | BiT-Basel. | LiT | 3T | Basel. | LiT | 3T |
|---|---|---|---|---|---|---|---|---|
| Retrieval | Flickr* img2txt | 71.2 | 72.5 | 61.1 | 74.6 | 71.2 | 60.4 | 74.0 |
| | Flickr txt2img | 49.3 | 50.6 | 39.1 | 52.5 | 49.3 | 35.0 | 52.1 |
| | Flickr* txt2img | 51.3 | 53.7 | 42.9 | 54.5 | 51.3 | 38.3 | 54.0 |
| | Flickr img2txt | 68.5 | 68.1 | 59.2 | 70.1 | 68.5 | 57.5 | 69.2 |
| | COCO img2txt | 44.3 | 45.0 | 40.9 | 46.8 | 44.3 | 38.9 | 45.7 |
| | COCO txt2img | 29.0 | 29.8 | 24.7 | 32.0 | 29.0 | 21.2 | 31.4 |
| **Average** | | 52.3 | 53.3 | 44.7 | 55.1 | 52.3 | 41.9 | 54.4 |
| Few-Shot Classification | Caltech | 89.8 | 90.0 | 90.4 | 90.9 | 89.8 | 91.0 | 92.1 |
| | UC Merced | 89.3 | 84.0 | 91.6 | 91.2 | 89.3 | 94.2 | 92.1 |
| | Cars | 75.0 | 73.4 | 42.4 | 77.1 | 75.0 | 49.4 | 78.7 |
| | DTD | 66.5 | 64.0 | 66.8 | 69.4 | 66.5 | 63.5 | 70.1 |
| | Col-Hist | 68.3 | 66.6 | 84.8 | 72.9 | 68.3 | 85.6 | 75.2 |
| | Birds | 44.5 | 38.5 | 85.5 | 53.2 | 44.5 | 62.1 | 52.5 |
| | Pets | 75.8 | 65.5 | 89.2 | 78.9 | 75.8 | 82.5 | 76.3 |
| | IN-1k | 52.0 | 57.8 | 73.8 | 56.2 | 52.0 | 60.4 | 56.4 |
| | CIFAR-100 | 57.8 | 40.0 | 74.1 | 62.1 | 57.8 | 62.3 | 61.0 |
| **Average** | | 68.8 | 64.4 | 77.6 | 72.4 | 68.8 | 72.3 | 72.7 |
| Zero-Shot Classification | Pets | 75.4 | 79.0 | 80.9 | 80.2 | 75.4 | 83.5 | 78.7 |
| | IN-1k | 59.8 | 60.6 | 67.1 | 62.8 | 59.8 | 62.9 | 62.5 |
| | CIFAR-100 | 60.0 | 45.0 | 72.6 | 59.6 | 60.0 | 56.0 | 61.1 |
| | IN-A | 32.8 | 31.4 | 26.2 | 34.9 | 32.8 | 21.7 | 32.5 |
| | IN-R | 75.4 | 73.4 | 55.4 | 78.1 | 75.4 | 57.0 | 77.4 |
| | IN-v2 | 53.1 | 53.3 | 58.8 | 55.7 | 53.1 | 55.0 | 55.6 |
| | ObjectNet | 44.7 | 42.7 | 34.4 | 47.3 | 44.7 | 25.3 | 45.0 |
| | Caltech | 79.1 | 81.0 | 76.7 | 79.3 | 79.1 | 79.0 | 80.7 |
| | DTD | 51.4 | 52.8 | 46.6 | 54.1 | 51.4 | 41.9 | 52.4 |
| | Eurosat | 36.5 | 19.7 | 28.1 | 32.6 | 36.5 | 32.6 | 36.2 |
| | Flowers | 51.9 | 54.3 | 64.1 | 57.1 | 51.9 | 48.2 | 55.6 |
| | RESISC | 53.1 | 49.9 | 27.8 | 52.9 | 53.1 | 24.4 | 48.0 |
| | Sun397 | 62.6 | 63.4 | 60.9 | 65.3 | 62.6 | 53.7 | 63.2 |
| **Average** | | 57.1 | 55.0 | 54.8 | 59.2 | 57.1 | 50.4 | 58.4 |
| **Average** | | 59.6 | 57.4 | 59.5 | 62.2 | 59.6 | 55.1 | 61.8 |

### A.6 IN-21k Pretraining – Additional Results

In Table A.4, we report retrieval and few-/zero-shot classification performance for 3T, LiT, and the baseline across a selection of pretraining datasets for L scale models and IN-21k pretraining. In addition to the unfiltered WebLI split reported in the main text, we here report results on the pair-filtered and text-filtered splits of the WebLI dataset, cf. §4.2. Further, we report results on the same dataset used by Zhai et al. [85] to train LiT. Across all datasets, 3T outperforms LiT and the baseline on average for retrieval and few-/zero-shot classification tasks, confirming our results for the unfiltered WebLI split in §4.1 and §4.2.

Table A.4: Results for the baseline, LiT, and 3T for L scale models and IN-21k pretraining. Across all datasets, 3T outperforms LiT and the baseline on average for retrieval, few-shot image classification, and zero-shot image classification tasks.

| Dataset Method | | Pair-Filtered WebLI Basel. | LiT | 3T | Text-Filtered WebLI Basel. | LiT | 3T | LiT Dataset Basel. | LiT | 3T |
|---|---|---|---|---|---|---|---|---|---|---|
| Retrieval | Flickr img2txt | 79.4 | 72.4 | 81.6 | 79.8 | 72.1 | 83.9 | 79.1 | 71.7 | 82.0 |
| | Flickr* img2txt | 81.0 | 73.5 | 83.8 | 84.5 | 75.8 | 86.9 | 80.2 | 74.5 | 84.2 |
| | Flickr txt2img | 59.6 | 48.3 | 62.5 | 65.1 | 51.3 | 68.8 | 60.6 | 48.1 | 64.1 |
| | Flickr* txt2img | 61.8 | 51.5 | 64.0 | 67.9 | 54.2 | 70.6 | 61.5 | 50.8 | 66.1 |
| | COCO img2txt | 56.4 | 47.8 | 58.4 | 58.2 | 50.3 | 62.5 | 53.0 | 47.2 | 57.6 |
| | COCO txt2img | 39.1 | 29.5 | 40.9 | 43.0 | 31.8 | 45.6 | 38.3 | 29.6 | 41.3 |
| **Average** | | 62.9 | 53.8 | 65.2 | 66.4 | 55.9 | 69.7 | 62.1 | 53.6 | 65.9 |
| Few-Shot Classification | IN-1k | 67.8 | 79.0 | 71.9 | 64.9 | 79.0 | 69.6 | 63.3 | 79.0 | 68.6 |
| | CIFAR-100 | 71.0 | 83.6 | 75.1 | 73.6 | 83.6 | 77.0 | 73.0 | 83.6 | 74.9 |
| | Caltech | 88.8 | 88.4 | 90.8 | 89.8 | 88.4 | 90.5 | 90.7 | 88.4 | 92.0 |
| | Pets | 91.3 | 89.2 | 91.7 | 85.7 | 89.2 | 87.7 | 84.1 | 89.2 | 86.9 |
| | DTD | 73.2 | 69.2 | 74.8 | 71.5 | 69.2 | 75.7 | 71.3 | 69.2 | 73.1 |
| | UC Merced | 94.3 | 92.8 | 96.1 | 94.8 | 92.8 | 95.8 | 92.8 | 92.8 | 94.6 |
| | Cars | 91.5 | 41.9 | 92.4 | 86.5 | 41.9 | 89.0 | 85.3 | 41.9 | 87.8 |
| | Col-Hist | 77.5 | 86.4 | 81.3 | 76.1 | 86.4 | 80.4 | 72.2 | 86.4 | 78.6 |
| | Birds | 70.9 | 83.4 | 79.0 | 56.7 | 83.4 | 68.2 | 62.1 | 83.4 | 70.4 |
| **Average** | | 80.7 | 79.3 | 83.7 | 77.7 | 79.3 | 81.6 | 77.2 | 79.3 | 80.8 |
| Zero-Shot Classification | IN-1k | 75.1 | 77.5 | 76.5 | 72.1 | 76.6 | 73.8 | 72.1 | 77.4 | 74.8 |
| | CIFAR-100 | 73.0 | 82.4 | 74.0 | 76.8 | 82.9 | 78.0 | 76.0 | 83.1 | 77.0 |
| | Caltech | 84.2 | 78.5 | 83.9 | 80.3 | 80.8 | 84.5 | 81.4 | 80.7 | 84.7 |
| | Pets | 93.5 | 91.6 | 94.4 | 87.7 | 88.3 | 90.8 | 86.5 | 88.4 | 88.6 |
| | DTD | 56.4 | 50.2 | 57.8 | 61.2 | 49.1 | 59.4 | 61.8 | 54.6 | 60.7 |
| | IN-A | 51.8 | 46.7 | 54.8 | 57.6 | 46.7 | 60.4 | 56.2 | 47.3 | 58.9 |
| | IN-R | 88.4 | 67.4 | 89.6 | 89.0 | 67.2 | 90.3 | 88.4 | 67.1 | 90.0 |
| | IN-v2 | 67.8 | 68.9 | 69.8 | 65.4 | 68.0 | 67.5 | 65.3 | 68.6 | 68.3 |
| | ObjectNet | 52.6 | 43.3 | 54.6 | 57.3 | 42.4 | 59.0 | 54.3 | 42.6 | 56.5 |
| | Eurosat | 37.2 | 28.5 | 44.2 | 38.2 | 20.2 | 42.4 | 44.0 | 29.1 | 45.8 |
| | Flowers | 79.7 | 81.1 | 80.3 | 63.8 | 77.8 | 67.7 | 67.3 | 79.0 | 73.4 |
| | Resisc | 61.6 | 32.5 | 63.1 | 61.8 | 30.0 | 63.1 | 58.3 | 28.4 | 57.2 |
| | Sun397 | 67.1 | 63.9 | 67.6 | 69.3 | 65.6 | 69.6 | 69.3 | 65.8 | 69.5 |
| **Average** | | 68.9 | 64.1 | 70.7 | 68.7 | 63.2 | 70.8 | 68.5 | 64.2 | 70.5 |
| **Average** | | 71.1 | 66.4 | 73.3 | 70.8 | 66.3 | 73.6 | 69.7 | 66.4 | 72.5 |

## A.7 JFT Pretraining – Additional Results

In Table A.5, we report few- and zero-shot classification performance for 3T, LiT, and the baseline across our selection of datasets for L scale models and JFT pretraining. LiT outperforms 3T and the baseline on average for few- and zero-shot classification tasks.

In Table A.6, we report performance for g scale models and JFT pretraining across all three splits of the WebLI dataset described in §4. Retrieval performance is generally best for all methods for the Text-Filtered WebLI split, with 3T generally performing best across splits and tasks. For classification, for 3T and the baseline, performance on Text- and Pair-Filtered WebLI is significantly better than on Unfiltered WebLI, with LiT generally performing best across splits. In line with our previous observations, the differences between the WebLI splits are smaller for LiT. As the image tower is kept fixed during contrastive training, LiT performance is influenced less by the contrastive learning setup.

**Retrieval Results: Comparison to SOTA.** While our retrieval performance is competitive, 3T does not set a new state-of-the art, see, for example, the CoCa paper [81] (Table 3) for a comparison of current methods. While SOTA results were never the aim of this paper—we instead study pretrained models for contrastive learning—there are a few advantages the CoCa setup has, and from which 3T would likely benefit, too. Most notably, CoCa trains for about 6 times more examples seen than we do here (32B vs. 5B). Our scaling experiments, cf. Fig. A.4, suggest we would expect a significant performance increase for longer training. There are further differences that likely benefit CoCa, such as the use of a larger batch size (65k for them vs 14k for us) or training on images with higher resolution for a portion of training (CoCa goes from 288×288 to 576×576, we stay at 288×288)—both of these changes significantly increase computational costs beyond the budget available to us: while CoCa training takes 'about 5 days on 2,048 CloudTPUv4 chips'[81], our g scale runs train for about the same duration on only 512 v4 TPU chips. It would be interesting to see if, in a fairer comparison, 3T matches or outperforms CoCa for retrieval tasks. Alternatively, ideas from 3T could also be used to improve CoCa-like architectures.

Table A.5: For JFT-pretraining, LiT outperforms 3T and the baseline on average on few- and zero-shot classification tasks. L scale models trained on Unfiltered WebLI.

| Method | | Basel. | LiT | 3T |
|---|---|---|---|---|
| Few-Shot Classification | IN-1k | 62.8 | 81.3 | 67.7 |
| | CIFAR-100 | 70.4 | 83.2 | 74.3 |
| | Caltech | 91.0 | 89.0 | 91.8 |
| | Pets | 85.9 | 96.8 | 88.4 |
| | DTD | 70.3 | 72.1 | 72.4 |
| | UC Merced | 91.8 | 95.5 | 93.1 |
| | Cars | 81.5 | 92.9 | 87.1 |
| | Col-Hist | 71.7 | 81.3 | 77.0 |
| | Birds | 53.4 | 85.6 | 62.4 |
| Zero-Shot Classification | IN-1k | 69.5 | 80.1 | 72.0 |
| | CIFAR-100 | 73.5 | 80.1 | 75.2 |
| | Caltech | 81.9 | 79.5 | 82.5 |
| | Pets | 84.2 | 96.3 | 88.7 |
| | DTD | 58.6 | 59.0 | 59.0 |
| | IN-C | 49.6 | 68.1 | 52.8 |
| | IN-A | 53.0 | 69.1 | 56.4 |
| | IN-R | 85.8 | 91.7 | 88.4 |
| | IN-v2 | 62.2 | 74.0 | 65.4 |
| | ObjectNet | 56.2 | 61.9 | 59.3 |
| | EuroSat | 32.7 | 36.6 | 54.7 |
| | Flowers | 62.0 | 76.7 | 66.6 |
| | RESISC | 58.0 | 58.9 | 60.9 |
| | Sun397 | 67.6 | 69.7 | 68.1 |
| **Average** | | 68.4 | 77.4 | 72.4 |

Table A.6: Results for the baseline, LiT, and 3T for g scale models and JFT pretraining for a selection of different splits of the WebLI dataset. 3T outperforms LiT for retrieval tasks, while LiT performs better for image classification. The from-scratch CLIP/ALIGN-style baseline is not competitive.

| Dataset Method | | Unfiltered WebLI | | | Pair-Filtered WebLI | | | Text-Filtered WebLI | | |
| --- | --- | --- | --- | --- | --- | --- | --- | --- | --- | --- |
| | | Basel. | LiT | 3T | Basel. | LiT | 3T | Basel. | LiT | 3T |
| Retrieval | Flickr img2txt | 75.2 | 83.0 | 81.5 | 81.4 | 83.2 | 84.0 | 85.0 | 83.9 | 87.3 |
| | Flickr* img2txt | 80.0 | 84.8 | 84.2 | 80.7 | 83.9 | 85.6 | 86.7 | 85.2 | 88.3 |
| | Flickr txt2img | 58.2 | 61.3 | 64.3 | 61.4 | 63.9 | 66.5 | 67.0 | 66.5 | 72.1 |
| | Flickr* txt2img | 60.1 | 63.1 | 65.6 | 62.7 | 65.4 | 68.4 | 68.2 | 67.6 | 72.9 |
| | COCO img2txt | 52.3 | 57.7 | 57.5 | 58.4 | 59.7 | 61.7 | 60.0 | 59.5 | 64.1 |
| | COCO txt2img | 37.5 | 40.0 | 41.1 | 41.2 | 41.9 | 43.9 | 44.7 | 43.6 | 48.5 |
| Few-Shot Classification | IN-1k | 67.5 | 84.6 | 72.8 | 71.8 | 84.6 | 75.7 | 69.6 | 84.6 | 73.9 |
| | CIFAR-100 | 72.7 | 83.2 | 78.0 | 73.1 | 83.2 | 78.7 | 76.4 | 83.2 | 80.0 |
| | Caltech | 91.8 | 90.0 | 93.3 | 89.7 | 90.0 | 90.9 | 90.8 | 90.0 | 92.4 |
| | Pets | 88.4 | 97.8 | 91.5 | 93.0 | 97.8 | 94.3 | 88.8 | 97.8 | 91.4 |
| | DTD | 70.7 | 74.6 | 74.7 | 74.2 | 74.6 | 76.1 | 73.6 | 74.6 | 76.0 |
| | UC Merced | 92.9 | 96.9 | 94.7 | 95.2 | 96.9 | 95.6 | 95.2 | 96.9 | 96.5 |
| | Cars | 84.1 | 93.3 | 88.6 | 92.6 | 93.3 | 93.5 | 89.0 | 93.3 | 91.6 |
| | Col-Hist | 72.0 | 83.6 | 76.2 | 77.8 | 83.6 | 80.9 | 73.5 | 83.6 | 79.4 |
| | Birds | 60.7 | 89.7 | 69.8 | 76.4 | 89.7 | 80.7 | 62.5 | 89.7 | 71.1 |
| Zero-Shot Classification | IN-1k | 73.5 | 84.0 | 76.3 | 78.0 | 84.7 | 79.6 | 75.8 | 84.3 | 78.2 |
| | CIFAR-100 | 77.5 | 81.3 | 80.3 | 76.2 | 81.3 | 79.5 | 80.6 | 81.8 | 82.3 |
| | Caltech | 79.8 | 81.4 | 82.3 | 84.0 | 82.4 | 82.9 | 79.5 | 80.9 | 81.9 |
| | Pets | 87.0 | 96.4 | 92.7 | 92.8 | 97.7 | 93.0 | 88.1 | 96.5 | 91.5 |
| | DTD | 59.2 | 62.1 | 64.9 | 58.9 | 55.6 | 60.1 | 61.4 | 62.0 | 62.1 |
| | IN-C | 54.9 | 72.9 | 58.2 | 57.7 | 73.3 | 60.3 | 57.6 | 73.3 | 61.3 |
| | IN-A | 64.9 | 80.2 | 67.8 | 59.9 | 79.5 | 65.1 | 67.8 | 80.5 | 70.8 |
| | IN-R | 89.8 | 94.4 | 91.8 | 90.5 | 94.2 | 92.8 | 91.8 | 94.6 | 93.3 |
| | IN-v2 | 66.4 | 78.1 | 69.5 | 70.8 | 79.2 | 73.0 | 69.1 | 78.5 | 71.4 |
| | Objectnet | 62.7 | 70.3 | 65.3 | 56.9 | 68.3 | 59.5 | 63.3 | 70.0 | 65.9 |
| | Eurosat | 55.7 | 33.6 | 48.9 | 32.9 | 30.7 | 42.8 | 47.9 | 36.1 | 52.1 |
| | Flowers | 71.0 | 84.2 | 73.5 | 82.4 | 86.3 | 83.0 | 69.4 | 86.6 | 72.5 |
| | RESISC | 61.5 | 58.4 | 60.5 | 59.8 | 56.5 | 64.8 | 65.4 | 57.8 | 61.7 |
| | Sun397 | 68.8 | 71.0 | 70.3 | 68.9 | 71.9 | 69.8 | 70.2 | 71.6 | 70.9 |
| **Average** | | 70.2 | 77.0 | 73.7 | 72.4 | 77.0 | 75.3 | 73.1 | 77.7 | 75.9 |

## A.8 Pretraining Robustness – Additional Results

In Table A.7, we report results on additional tasks for 3T, LiT, and the baseline for both the 'mismatched' setup and Places365 pretraining of §4.4. We find again that 3T is much more robust in both setups, significantly outperforming LiT. The difference is particularly striking when using models pretrained on Places365, where LiT's performance degrades drastically while 3T is still able to improve over the baseline.

Table A.7: Testing robustness to the 'mismatched setup' and Places365 pretraining (instead if IN-21k/JFT) for 3T and LiT. In both cases, 3T performs significantly better than LiT. In particular when using models pretrained on Places365, LiT's performance degrades dramatically while 3T continues to improve over the baseline on average. (Note that the baselines here are different not because they use the pretraining dataset, but because we compare to an L scale baseline for the mismatched setup and a B scale baseline (trained for only 900M examples seen) for Places365 pretraining.) We refer to the main text for full details.

| Experiment | Method | Mismatched Setup | | | Places365 Pretraining | | |
|---|---|---|---|---|---|---|---|
| | | Basel. | LiT | 3T | Basel. | LiT | 3T |
| Retrieval | Flickr* img2txt | 75.6 | 66.5 | 80.2 | 56.0 | 35.5 | 58.1 |
| | Flickr* txt2img | 57.1 | 45.1 | 62.1 | 36.2 | 19.5 | 38.4 |
| | COCO img2txt | 51.0 | 44.1 | 54.5 | 34.1 | 19.3 | 36.5 |
| | COCO txt2img | 34.2 | 26.4 | 37.8 | 21.0 | 10.9 | 22.1 |
| Few-Shot Classification | IN-1k | 62.8 | 70.3 | 67.6 | 37.8 | 16.6 | 41.5 |
| | CIFAR-100 | 70.4 | 80.3 | 73.8 | 47.1 | 33.9 | 52.7 |
| | Caltech | 91.0 | 88.1 | 91.7 | 87.9 | 66.5 | 88.5 |
| | Pets | 85.9 | 86.0 | 86.8 | 56.8 | 20.3 | 59.9 |
| | DTD | 70.3 | 66.3 | 73.4 | 58.4 | 39.7 | 63.1 |
| | UC Merced | 91.8 | 91.5 | 93.8 | 85.8 | 80.8 | 89.4 |
| | Cars | 81.5 | 36.7 | 85.3 | 57.0 | 10.1 | 58.6 |
| | Col-Hist | 71.7 | 84.4 | 74.3 | 72.9 | 70.7 | 78.7 |
| | Birds | 53.4 | 76.8 | 65.2 | 33.2 | 15.7 | 38.1 |
| Zero-Shot Classification | IN-1k | 69.5 | 69.5 | 71.5 | 45.6 | 24.5 | 47.4 |
| | CIFAR-100 | 73.5 | 78.6 | 75.6 | 48.3 | 27.4 | 52.4 |
| | Caltech | 81.9 | 82.0 | 81.2 | 76.6 | 62.7 | 77.0 |
| | Pets | 84.2 | 84.7 | 87.4 | 61.5 | 30.3 | 60.2 |
| | DTD | 58.6 | 49.4 | 60.6 | 39.8 | 23.6 | 39.7 |
| | IN-C | 49.6 | 55.5 | 51.8 | 25.3 | 14.4 | 27.3 |
| | IN-A | 53.0 | 29.1 | 54.1 | 12.0 | 4.7 | 12.5 |
| | IN-R | 85.8 | 60.7 | 87.9 | 56.1 | 20.3 | 58.2 |
| | IN-v2 | 62.2 | 61.1 | 65.0 | 39.4 | 20.7 | 40.5 |
| | Objectnet | 56.2 | 34.9 | 57.8 | 28.4 | 7.3 | 29.6 |
| | Eurosat | 32.7 | 33.1 | 52.5 | 33.7 | 15.6 | 27.3 |
| | Flowers | 62.0 | 74.1 | 66.2 | 37.6 | 17.4 | 37.3 |
| | RESISC | 58.0 | 29.0 | 57.4 | 37.9 | 24.0 | 38.3 |
| | Sun397 | 67.6 | 62.0 | 68.4 | 55.1 | 60.6 | 57.3 |
| **Average** | | 66.4 | 61.7 | 69.8 | 47.5 | 29.4 | 49.3 |

## A.9 Scaling Model Sizes and Training Duration – Additional Results

Complementing the results of §4.3, in Fig. A.4 we report the performance when scaling only the number of examples seen during training, keeping the model sizes fixed at B scale. We observe a similar trend to §4.3 / Fig. 3, where 3T benefits more from increases in scale than LiT. Compared to the baseline, 3T consistently improves performance, even as the number of examples grows very large. We do not observe any evidence that 3T only improves performance for particular compute budgets. Note that, because the dataset size is 10B samples, all of our runs equate to less than a full epoch.

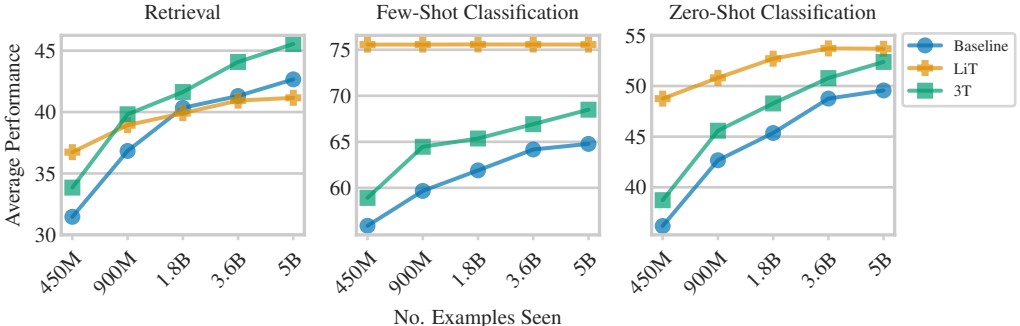

Figure A.4: Increasing training duration of 3T, LiT, and the baseline; average retrieval, few- and zero-shot classification performance. The model scale is B (B/32 for ViTs) for all approaches and towers. 3T and the baseline benefit more from increases in scale than LiT, with 3T maintaining a consistent increase in performance over the baseline. Note that the few-shot performance for LiT is fixed, as only the locked pretrained image tower is used for fewshot applications.

## A.10 Benefits From Using 3T With All Three Towers at Test Time – Extended Version

We usually discard the pretrained model when applying 3T to downstream tasks, cf. Fig. 2 (b). Instead, in this section, we explore whether we can find benefits from using the locked third tower at test time, similar to LiT. More specifically, we are interested in interpolating between the main image tower and locked pretrained model in the third tower. Can we interpolate between 3T- and LiT-like prediction by combining the image embeddings?

This idea does not work directly with the default 3T due to our use of linear projection heads, cf. Fig. 2 (a), since there is no unified embedding space that all towers embed to. Therefore, we introduce a 'headless 3T' variant, for which we do not use the linear projection heads, $h_f$, $h_g$, $f_h$, and $g_h$. (Alternatively, one may think of all linear projection heads fixed to identity mappings.) Thus, all losses directly use the same embeddings, $f(I)$, $p(I)$, and $g(T)$, making the embedding spaces directly comparable. Here, we train B scale models for 3.6B examples seen and use an IN-21k-pretrained model. Further note that the average zero-shot classification performance we report here is over only a subset of the list of tasks used in §4.2: we consider IN-1k, CIFAR-100, and Pets. The selection of few-shot classification and retrieval tasks remains the same, although we do not use the Karpathy split for Flickr here.[*]

In Fig. A.5, we display the average retrieval, few-shot classification, and zero-shot classification performance for the convex combination, alongside a comparable LiT run and a 3T run with default projection head setup. Across all tasks, we observe similar behavior: for $\alpha = 0$ (full weight on the third tower), we obtain performance close to, but ultimately below, LiT; performance then increases with $\alpha$, peaking for $\alpha \in [1/4, 3/4]$, before decreasing again. At $\alpha = 1$ (full weight on main image tower), we recover the performance of the headless 3T setup. Interestingly, for retrieval and few-shot classification tasks, the convex combination yields better performance than either of the towers separately across a relatively broad band of $\alpha$ values.

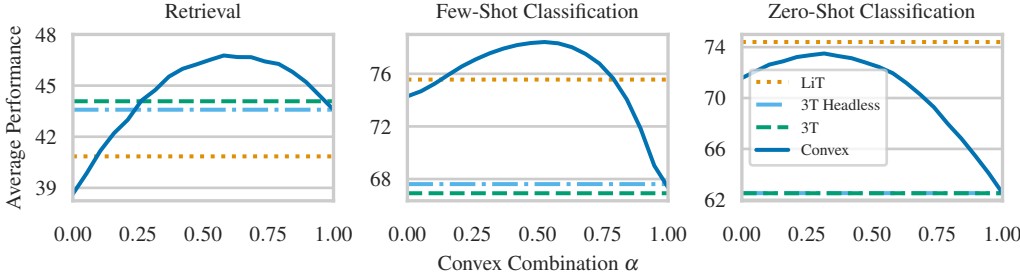

Figure A.5: Convex combination of the image models in 3T: $\alpha \cdot h(I) + (1-\alpha) \cdot f(I)$. By varying $\alpha$, we can generally interpolate between 3T and LiT performance. Interestingly, for a broad range of weights, the retrieval and few-shot classification performance of the combination outperforms 3T and LiT.

Perhaps counterintuitively, for $\alpha = 0$, we do not recover the performance of LiT exactly. The reasons for this differ between tasks: For retrieval and zero-shot applications, while the image tower is identical to that of LiT, the text tower is different as it has been trained with the 3T objective. For few-shot application, the default evaluation procedure of Zhai et al. [85] uses the prelogits of the ViTs underlying $f$ and $h$ as inputs to the few-shot classifier, i.e. not the final embeddings. As the prelogit spaces of $f$ and $h$ are not aligned, here, we need to instead construct the convex combination in embedding space, which does however mean that $\alpha = 0$ does not give performance equivalent to LiT. Lastly, although the 3T run with the default projection heads does not seem to perform better than '3T headless' in this instance, we have seen 'headless' setups underperform in preliminary experiments and would suggest additional experiments before opting for a headless design, see also §A.11.

We believe that further study of this approach is exciting future work: the method is entirely post-hoc and no additional training costs are incurred, although inference costs do increase.

## A.11   Ablation

In this section, we give additional results and details for the ablation study presented in §4.6. Table A.8 gives additional results, extending Table 5 in the main paper. In addition to the mean and two standard errors, we also report standard deviations over tasks here. Note that, for zero-shot classification performance, we only have access to a subset of the full list of tasks used in Section 4.2: we consider IN-1k, CIFAR-100, and Pets. The selection of few-shot classification and retrieval tasks remains the same, although we do not use the Karpathy split for Flickr here.[*]

**No $\mathcal{L}_{\cdot \leftrightarrow \cdot}$. – Details.** For this ablation we consider leaving out either of the three loss terms. 'No $\mathcal{L}_{f \leftrightarrow g}$': We replace the 3T loss by $\frac{1}{2} \cdot (\mathcal{L}_{f_h \leftrightarrow h_f} + \mathcal{L}_{g_h \leftrightarrow h_g})$. 'No $\mathcal{L}_{f_h \leftrightarrow h_f}$': We replace the 3T loss by $\frac{1}{2} \cdot (\mathcal{L}_{f \leftrightarrow g} + \mathcal{L}_{g_h \leftrightarrow h_g})$. 'No $\mathcal{L}_{g_h \leftrightarrow h_g}$': We replace the 3T loss by $\frac{1}{2} \cdot (\mathcal{L}_{f \leftrightarrow g} + \mathcal{L}_{f_h \leftrightarrow h_f})$. When leaving out either of the three loss terms, average performance suffers significantly. Leaving out the loss between the main two towers (obviously) has the biggest negative effect, as the main embeddings, $f(I)$ and $g(T)$, are not aligned during training.

**Head Variants – Details and Additional Results.** In the main part of the paper, we have only given results for the best alternative variant for the projection head setup. Here, we describe all variants and report results individually. We refer to Fig. 2 (a) for the projection head notation. 'Heads only on Third Tower': The main tower projection heads $f_h$ and $g_h$ are fixed to identity mappings. 'Heads Only on Main Towers': The third tower projection heads $h_f$ and $h_g$ are fixed to identity mappings. 'No Heads/Headless': This is the setup described in §A.10: all linear projections $h_f, h_g, f_h, g_h$ are fixed to identity mappings. 'Heads Fully Independent': This setup adds linear projection heads before the computation of $\mathcal{L}_{f \leftrightarrow g}$, i.e. we compute $f_g(I) = \texttt{Lin}(f(I))$ and $g_f(T) = \texttt{Lin}(g(T))$, and then compute the loss $\mathcal{L}_{f_g \leftrightarrow g_f}$ (instead of $\mathcal{L}_{f \leftrightarrow g}$). In Table A.8, we give results for all variants that we try; none outperform the base variant significantly, while some underperform.

**MLP Embedding – Details.** When replacing the linear projection $h$ in the third tower with an MLP, we use the following architecture: $\texttt{MLP}(x) = \texttt{Lin}_2(\texttt{GELU}(\texttt{Lin}_1(x))$, where we use GELU non-linearities [29], $\texttt{Lin}_1$ expands the embedding dimensionality of the input by a factor of $4$, and $\texttt{Lin}_2$ maps to the shared embedding dimension $D$.

**3T with Loss Weights – Details and Additional Results**. We replace the standard 3T loss with a weighted objective $\frac{1}{3} \cdot (\mathcal{L}_{f \leftrightarrow g} + w \cdot (\mathcal{L}_{f_h \leftrightarrow h_f} + \mathcal{L}_{g_h \leftrightarrow h_g}))$. For the weights $w$, we sweep over

Table A.8: Extended results for the 3T ablation study. Difference to the 3T reference run for various architecture ablations. We report mean, standard deviation, and two standard errors of the differences over the downstream task selection.

| Difference | Mean | Standard Deviation | Two Standard Errors |
|---|---|---|---|
| Rerun | -0.22 | 0.50 | 0.25 |
| No $\mathcal{L}_{f \leftrightarrow g}$ | -26.63 | 21.22 | 10.61 |
| No $\mathcal{L}_{f_h \leftrightarrow h_f}$ | -1.19 | 1.51 | 0.75 |
| No $\mathcal{L}_{g_h \leftrightarrow h_g}$ | -2.77 | 1.83 | 0.91 |
| (Head Variants (best)) | 0.09 | 0.70 | 0.35 |
| Heads Only on Third Tower | 0.09 | 0.70 | 0.35 |
| Heads Only on Main Towers | -0.67 | 0.66 | 0.33 |
| Heads Fully Independent | -0.60 | 0.63 | 0.32 |
| No Heads/Headless | -0.47 | 1.04 | 0.52 |
| MLP Embedding | -0.08 | 0.69 | 0.35 |
| More Temperatures | -0.26 | 0.95 | 0.48 |
| (Loss weight = 2 (best)) | 0.17 | 1.06 | 0.53 |
| Loss weight 0.1 | -2.31 | 1.33 | 0.67 |
| Loss weight 0.5 | -0.90 | 0.81 | 0.41 |
| Loss weight 2 | 0.17 | 1.06 | 0.53 |
| Loss weight 10 | -0.56 | 1.74 | 0.87 |
| (L2 Transfer (best)) | -3.80 | 2.27 | 1.13 |
| L2 Transfer w=0.0001 | -4.40 | 1.89 | 0.94 |
| L2 Transfer w=0.001 | -3.80 | 2.27 | 1.13 |
| L2 Transfer w=0.05 | -4.41 | 2.24 | 1.12 |
| L2 Transfer w=0.01 | -4.17 | 1.97 | 0.99 |
| L2 Transfer w=.1 | -3.97 | 2.06 | 1.03 |
| L2 Transfer w=.5 | -7.12 | 2.95 | 1.48 |
| L2 Transfer w=1 | -11.38 | 4.39 | 2.19 |
| L2 Transfer w=2 | -16.09 | 5.14 | 2.57 |
| L2 Transfer w=10 | -46.80 | 14.32 | 7.16 |
| 3T Finetuning | 1.85 | 2.53 | 1.27 |

$w \in \{0.1, 0.5, 2, 10\}$. All weights except $w = 2$ lead to an average performance decrease. However, the size of the effect for $w = 2$ is small relative to twice the standard error.

**L2 Representation Transfer – Details and Additional Results.** We investigate the use of squared losses for the representation transfer between the main towers and the third tower instead of relying on the contrastive loss. Concretely, we replace the 3T loss, Eq. (4), with

$$\frac{1}{3} \left\{ \mathcal{L}_{f \leftrightarrow g} + w \frac{1}{N} \sum_{i=i}^{N} \left[ \| f_h(I_i) - h_f(I_i) \|^2 + \| g_h(T_i) - h_g(I_i) \|^2 \right] \right\}. \tag{5}$$

For the weight hyperparameters $w$, we sweep over a large set of values, $w \in \{0.0001, 0.001, 0.05, 0.01, 0.1, 0.5, 1, 2, 10\}$. L2 representation transfer gives worse results than the contrastive loss for all values of $w$ we try, corroborating the results of Tian et al. [71].

**Finetuning – Details and Additional Results.** Initializing the main tower in 3T with the same JFT-pretrained model as the third tower boosts performance significantly, increasing average performance from $56.76$ to $58.61$. A rerun confirmed these results; we obtained an increase from $56.46$ to $58.82$. Excited by this, we explored the 3T finetuning setup at other scales, and report performance in Table A.9. Note that here, we increase the numbers of examples seen during training from 450M (S scale) to 900M (B scale) to 5B (L scale). We observe that, as we increase the scale of the experiments, the gains from finetuning the main image tower decrease until they are negligible (compared to rerun variance). We therefore have opted to not make finetuning the main tower part of the standard 3T setup, as it (a) complicates the setup and (b) restricts the main tower to be the same model architecture and scale as the third tower.

Table A.9: Finetuning for 3T: Initializing the main tower in 3T with the same pretrained model as the third tower improves performance significantly at smaller but not larger experiment scales.

| Pretraining | Scale | Avg. Performance 3T | Avg. Performance 3T Finetuned |
|---|---|---|---|
| JFT | B | 56.76 | 58.61 |
| | L | 73.97 | 74.22 |
| IN-21k | S | 44.39 | 47.61 |
| | B | 56.30 | 58.83 |
| | L | 73.63 | 73.83 |

**'FlexiLiT 1/2' – Details.** With the FlexiLiT variants, cf. Table 5 in the main body of the paper, we investigate if there are other, simple ways to improve LiT. For both variants, we create a new 'half-locked' image tower by adding learnable components to the frozen pretrained image model. For FlexiLiT 1, we add a lightweight learnable 4-layer MLP on top of the frozen backbone: `FlexiLiT-1`$(I) = $ `MLP(LiT`$(I))$. The MLP has 4 layers, uses GELU-nonlinearities, and an expansion factor of 4. For FlexiLiT 2, we add an additional learnable ViT next to the locked backbone (adding significant cost) and merge representations with an MLP: `FlexiLiT-2`$(I) = $ `MLP(concat(LiT`$(I))$, `ViT`$(I))$. The additional ViT is B/32, following the main locked image tower. The MLP merging the two representations is an MLP with the same configuration as for FlexiLiT 1.

# B    Implementation Details

We follow Zhai et al. [85] for optimization and implementation details. We use the open-source vision transformer implementation available from Beyer et al. [4].

Unless otherwise mentioned, we use Transformers of scale L, with a $16{\times}16$ patch size for the ViT image towers, i.e. L/16. We train for 5B examples seen at a batch size of $14 \cdot 1024$, i.e. for about $350\,000$ steps. We resize input images to $224 \times 224$ resolution, and normalize pixel values to the $[-1, 1]$ range. Note that for experiments with g scale models, we resize images to $288 \times 288$ instead. We use a learning rate of $0.001$, warming up linearly for $10\,000$ steps, before following a cosine decay schedule. We use the Adafactor optimizer [65] with default $\beta_1 = 0.9$ and $\beta_2 = 0.99$, and we clip gradients if their norm exceeds $1.0$. For g scale runs, we set $\beta_2 = 0.95$ by default, which we found to be important to ensure training stability. We use weight decay of $0.001$.

We aggregate embeddings across tokens using multihead attention pooling, i.e. an attention block where the query is a single learned latent vector, and the keys and values are the outputs of the vision transformer (cf. `vit.py` in the code base [4]).

For details on how the different model scales and patch sizes relate to transformer width, depth, MLP dimension, the number of heads, or parameter count, we refer to Table 1 in [17] and Table 2 in [84].

**Compute Cost.** We train our models on v3 and v4 TPUs. For our main experiments at L scale, we use 256 TPU chips per experiment. Our 3T runs converge in about three days, for example, the 3T run with JFT pretraining took 63 hours of training time to converge over 348772 training steps. The baseline converges in 54 hours, and LiT in 35. For our five main experiments at L scale—3T, LiT for JFT and IN-21k pretraining, and a baseline run—the total runtime was about 280 hours, or about 8 TPU–Chip years worth of compute for the L scale experiments of this project. At g scale, we use 512 TPU chips per run, and our 3T runs converge in about 5 days.

Below we mention additional details pertaining to only some of the experiments.

**Details on Few-Shot Classification.** Following Zhai et al. [85], we use the prelogits of the ViTs instead of the final embeddings as input to the linear few-shot classifier.

**Details on Places Experiment.** Following Zhai et al. [85], for the Places365 experiment, we use a B/16 ResFormer [70] as the pretrained model.

## C  Societal Impact

With 3T, we introduce a novel machine learning method for learning joint embeddings of images and text. We train on large datasets of noisy and potentially biased data crawled from the internet. The same general caveats that apply to CLIP/ALIGN and LiT may also apply to 3T. We refer to §7 in Radford et al. [58] for a general discussion of the societal impact these methods may have.

Additionally, we wish to highlight the importance of carefully evaluating these models, testing for specific undesired behavior, before applying them in production. While the zero- and few-shot classification capabilities of these models are generally impressive, it is also important to consider their limitations and not succumb to wishful thinking when it comes to the real-world performance of these models on arbitrary tasks. For example, all of the approaches we study here do not perform well for zero-shot prediction on the structured and specialized tasks contained in VTAB, which include, for example, medical applications. It is therefore particularly important to carefully evaluate the performance of these methods when applied to real-world applications. Lastly, because 3T and LiT rely on two datasets for training, a classification and a contrastive learning dataset, this can complicate investigations into undesired biases in the final model.

## D  Libraries & Dataset

We rely on the Jax [5], Flax [26], and TensorFlow [1] Python libraries for our implementation. Additionally, we make use of the Big Vision [4] and Robustness Metrics [15] code bases.

For retrieval performance, we evaluate on Microsoft COCO [10] and Flickr30k [56]. For image classification, we evaluate on IN-1k [40, 61], CIFAR-100 [40], Caltech-256 [23], Oxford-IIIT Pet [53], Describable Textures (DTD) [13], UC Merced Land Use [80], Stanford Cars [39], Col-Hist [37], Birds [73], ImageNet variants -C [28], -A [32], -R [31], -v2 [59], ObjectNet [3], EuroSat [27], Oxford Flowers-102 [50], NWPU-RESISC45 [12], and Sun397 [78].

We take the EuroSat, Flowers, RESISC, and Sun397 datasets from the Visual Task Adaptation Benchmark (VTAB) [83]. They are the only VTAB datasets for which at least one method achieved better than trivial performance.

