# OpenReview forum: "Three Towers: Flexible Contrastive Learning with Pretrained Image Models"
_NeurIPS.cc/2023/Conference — NeurIPS 2023 poster_

### Official Review · Reviewer_Y95u · 2023-07-04

**Soundness:** 3 good
**Presentation:** 4 excellent
**Contribution:** 2 fair
**Rating:** 5
**Confidence:** 5

**Summary:**

CLIP is trained using contrastive loss. Previous works have shown that freezing the image tower, and only training the text encoder on pretraining datasets yields better performance compared to training both the encoders. This paper proposes that completely freezing the image encoder is drastic and will underperform when the downstream data is not covered in pretraining data span for the image tower.

Hence they propose training both the image and text encoder from scratch (as in standard training), albeit with an additional contrastive loss with the pretrained representations for the image encoder. They show that this approach outperforms the baseline standard CLIP training as well as LiT.

 However, it is not clear if the gains of the proposed approach will be seen as the pretraining image-caption corpus data size is increased, especially to the point at which the baseline has a similar zeroshot accuracy as JFT/Im21k pretrained representations.


**Strengths:**

Pros:
- The empirical gains are good, especially in retrieval tasks in the scales and settings considered.
- The idea is simple to implement and well described.
- Proper ablations have been done.


**Weaknesses:**

Cons:
- The novelty is limited. It is a well-established fact that distilling knowledge from pretrained models improves performance.
- LiT had the additional advantage of being cost-effective, whereas this will not be true for the proposed approach 3T. However, it is not a major concern or the goal of this work and will not be part of my evaluation. Probably, it would be a good idea to tone down the connection to LiT in the paper and not make it feel like a build-up over LiT or else make it clear early on that we do not want to be cost effective.
- Section 4.3 Scaling pretraining data: It would be much better to consider showing gains as the pretraining image-caption corpus is scaled i.e. LAION400M to 5B to 12B or so based on your compute budgets. More importantly, as the image-caption corpus is scaled, intuitively it seems that there will be diminishing returns from using the 3rd tower of pretrained image representations from say Im21k or JFT. Can authors maybe do some mini experiments where training samples are scaled from say 2M to 64M exponentially (5-6 points) or whatever scale is feasible for them (I believe this is quite small a scale given other experiments in the paper).
- As the pretraining image-caption scales are increased to 5B and 12B(https://www.datacomp.ai/leaderboard.html), the zeroshot accuracies have approached near 80% similar to that of JFT (81%, which is a proprietary dataset moreover) or higher than that of Im21k (79%). Would using 3T based training with the JFT/Im21k pretrained image (3rd tower) tower help here? If not, then does it suggest that gains when training on WebLI was just because of additional data seen by JFT or Im21k, the advantage which doesn’t hold as these image-caption corpuses are being scaled.
- However, at the same, it would be interesting if you can show that 3T can help improving the zeroshot accuracies of a smaller CLIP model if the representations from the bigger CLIP model are used. This is because such an observation might hold true even at large image-caption corpus scales.
- Beyond the fact that distillation, in general, helps improve the quality of learned representations, do authors have any other intuition behind why 3T is better than the baseline (not LiT, that is clear).


**Questions:**

See the cons section.

**Limitations:**

Yes

---

> ### Author Rebuttal · Authors · 2023-08-09
>
> Dear reviewer Y95u,
>
>
> Thank you for your hard work and helpful feedback. We look forward to actively engaging with you during the discussion period to clarify any remaining points. Please read our comment above, addressed to all reviewers, first.
>
>
> > Section 4.3 Scaling pretraining data [...]
>
>
> Thanks for raising this point! Our experiments in Appendix 3 may address your concerns already. In A.3, we increase the training set size from 450M to 5B samples.
> Our results indicate that gains from 3T over the CLIP/ALIGN baseline are constant across dataset sizes, and that 3T benefits more from larger datasets than LiT.
>
>
> We agree that it _could_ be possible, that at very large contrastive learning dataset sizes, the benefits from 3T over the baseline _could_ grow smaller as contrastive learning outweighs benefits from the pretrained model.
> However, we have not observed this, despite training on _very large_ datasets already.
> We also observe benefits for 3T when the pre-trained model is weak (see both experiments in S. 4.4), which is another setting in which one would perhaps expect 3T to not gain much over the CLIP baseline.
>
>
> Again, thanks for bringing this up and we will include this discussion in the paper, as soon as we are able to update it.
>
>
> > As the pretraining image-caption scales are increased [...]
>
>
> Our results indicate that 3T will still improve performance here.
> In Table A.2, we show performance for g scale models trained on the Pair-Filtered WebLi split.
> For the baseline, we obtain 78.0% zero-shot on IN-1k (our best baseline result and close to the JFT performance you cite).
> Despite this, 3T still improves IN-1k performance over the baseline with 79.6% accuracy.
>
>
> Another example are tasks such as Cars, DTD, IN-A, IN-R, ObjectNet, or EuroSat in Table 3: despite _the baseline outperforming LiT_, 3T still benefits from the pre-trained model, and _3T outperforms both the baseline and LiT_.
> See also our robustness results (S. 4.4 and discussion below), where we also observe 3T performing best, despite the baseline outperforming LiT.
>
>
> We are therefore confident that, even when the baseline is as good or better than the pre-trained model, 3T can improve performance over both.
>
>
> > However, at the same, it would be interesting if you can show [...]
>
>
> Thanks for this suggestion! We agree that distilling larger CLIP models into smaller CLIP models is interesting future work that deserves careful attention on its own.
>
>
> Our study of the ‘mismatched setup’ in S. 4.4 might get closest to answering the question of whether 3T will continue to be beneficial when contrastive learning dataset size increases further.
> Instead of increasing the contrastive learning dataset scale, we here _decrease_ the capabilities of the pre-trained model instead, by using a smaller ViT for the pre-trained model than for the main image tower.
> Predictably, we now observe that the baseline often outperforms LiT (cf. Table A.3).
> However, despite this, 3T continues to provide benefits over the baseline (and LiT).
>
>
> Together with our dataset scaling results (S. A.3, see discussion above), we are therefore confident that 3T would continue to provide benefits over both the baseline and LiT at larger dataset sizes.
>
>
> > The novelty is limited [...]
>
>
> Firstly, we agree the idea behind 3T is intuitive.
> However, our ablation (S. 4.6, A. 5) shows that actually obtaining benefits from distillation is not as simple: naive approaches, e.g. standard L2 losses, actually do not perform well, and we require _contrastive_ distillation objectives to reap benefits.
>
>
> Secondly, we further think it is an advantage that the final 3T architecture is straight-forward, not a disadvantage: 3T is easy to implement and provides consistent and convincing performance improvements.
> The NeurIPS reviewer guidelines explicitly mention that a ‘novel combination of well-known techniques’ can be ‘valuable’.
> This is at least what we provide by combining contrastive learning jointly with distillation, providing value by improving the performance of CLIP-style contrastive learning.
> Our results are not found in the existing literature: the primary competing prior work, LiT, uses pretrained models in a totally different fashion.
>
>
> > Beyond the fact that distillation [...]
>
>
> We suspect that distillation is the main mechanism behind why 3T improves performance over the baseline, and would be curious to hear if you have any reason to think otherwise.
> Unlike 3T, the baseline does not include knowledge from a pre-trained model.
> If that prior knowledge is incorporated appropriately, 3T should perform at least as good as the baseline – which is exactly what we usually observe.
> We find the following example from Table 3 in S. 4.2 illustrative:
> IN-21k contains a large variety of labels of bird species, and therefore, the IN-21k-pretrained model is much more knowledgeable about birds than the baseline.
> With 3T, we can incorporate this knowledge from the pretrained model about birds, and significantly outperform the baseline model on the birds task.
>
>
> As discussed, sometimes 3T improves over the baseline despite LiT performing worse than the baseline.
> On one hand, this can still be explained by distillation: regardless of performing worse, the pretrained model may contain information about the task that the baseline does not.
> On the other hand, there could also be more indirect benefits from aligning the image embedding space to the pre-trained classifier: there might be a general mismatch between the contrastive learning objective and downstream tasks that is ‘corrected’ by also aligning the towers to the classification embeddings.
>
>
> If you think this discussion is helpful, we would be more than happy to add it to the revised version of the paper.
>
>
> Again, thank you very much for your review. We hope that our reply has addressed your concerns. Please let us know if there are any further changes you would like to see or if there is anything else that we can clarify.

---

> > ### Comment · Reviewer_Y95u · 2023-08-13
> > **Discussion**
> >
> > I thank the authors for the clarifications.
> > Thanks for pointing to the large scale (5B) experiments in the appendix. I would encourage the authors to add that in the main paper.
> > Can the authors also shed some light on why the proposed approach should help at large scale when the baseline anyways has an accuracy near to the pretrained third tower?
> > I increase my rating to borderline accept.

---

> > > ### Author Response · Authors · 2023-08-18
> > > **Author Response**
> > >
> > > Thank you for your careful evaluation of our rebuttal. We are delighted that you feel our reply addresses your concerns well. We will happily add the data scaling experiments from Appendix 3 to the main paper for the next revision.
> > >
> > >
> > > > Can the authors also shed some light on why the proposed approach should help at large scale when the baseline anyways has an accuracy near to the pretrained third tower?
> > >
> > >
> > > Thanks for bringing this up and apologies for our delayed reply: we did not get a notification when you edited your initial response to include a question.
> > > First we’d like to emphasize that 3T _does_ often help in these situations; see our discussion on Table A.2 in our reply to you above.
> > >
> > >
> > > To your main point, we believe that 3T improves accuracy in the scenario you mention because of an implicit ensembling effect.
> > > Our distillation objective combines two different predictive mechanisms into a single image tower: we distill a classifier _and_ optimize the main contrastive learning objective at the same time.
> > > Importantly, the pre-trained model and the baseline may predict quite differently on particular test inputs, even if their global accuracies are the same – crucially, they might make their mistakes on different downstream task examples.
> > > By correctly combining their predictions, and thus reducing the number of mistakes in the combined prediction, one can obtain increased global accuracy, despite the accuracy of the individual models being the same.
> > > To demonstrate this, we are currently re-evaluating the g scale models trained on the Pair-Filtered WebLi split from Table A.2, and we are committed to getting these results to you as soon as possible.
> > >
> > >
> > > In the mean time, in some sense, we can also see an example of this in Table 3.
> > > The average accuracies across tasks for the Baseline (68.4%) and LiT (68.3%) are very similar.
> > > Yet, 3T obtains an improved average performance of 71.4%.
> > > Looking only at the aggregate performance across datasets hides the fact that the Baseline and LiT perform quite differently on individual datasets: for example, the Baseline performs poorly on tasks such as IN-1k, while LiT performs poorly on tasks such as Cars.
> > > Therefore, 3T can benefit from combining knowledge from the pre-trained model and contrastive learning, and obtains improved average accuracy across datasets.
> > > What we see happening in Table 3 on an ‘across-dataset’ level is likely also happening within individual datasets, where the Baseline and LiT perform well or bad for different inputs (perhaps belonging to particular classes), such that 3T benefits from combining knowledge from both the pre-trained model and contrastive learning even if the pre-trained model and baseline perform similarly on aggregate metrics.
> > >
> > >
> > > Lastly, we note that precedent for ‘a combination of similar strength models is better than any individual model’ is given by literature on ensembling methods [B].
> > > For example, deep ensembles [A] average the predictions of independently trained neural networks, which typically vary only in the random seed used for initializing the weights and the order of the data seen during training.
> > > As such, the accuracies of the individual ensemble members are usually very similar.
> > > However, combining their predictions consistently improves accuracies.
> > > Note that, moving closer to our approach, deep ensembles can also be distilled into a single model, cf., e.g. [C].
> > > With 3T, we observe benefits from combining the pre-trained model with the baseline contrastive learning objective for the reasons discussed above.
> > >
> > >
> > > We will happily add the above discussion and evaluation to the main paper.
> > > If you think the additional discussion here further strengthens our submission, we would be interested to help clear up any other concerns that perhaps stand in the way of you increasing your score further.
> > >
> > >
> > > [A]: Lakshminarayanan, Balaji, Alexander Pritzel, and Charles Blundell. "Simple and scalable predictive uncertainty estimation using deep ensembles." Advances in neural information processing systems 30 (2017).
> > >
> > > [B]: Breiman, Leo. "Random forests." Machine learning 45 (2001): 5-32.
> > >
> > > [C]: Nam, Giung, et al. "Diversity matters when learning from ensembles." Advances in neural information processing systems 34 (2021): 8367-8377.

---

> > > > ### Author Response · Authors · 2023-08-20
> > > > **Investigation Into Individual Predictions**
> > > >
> > > > Dear reviewer Y95u,
> > > >
> > > >
> > > > we are now able to provide you with a detailed evaluation of how predictions differ between 3T, LiT, and the baseline on a per datapoint (and per task) level.
> > > > We believe this evaluation gives meaningful insights into understanding how 3T improves predictions over LiT and the baseline.
> > > > Concretely, this data shows that the contrastive learning baseline and the pre-trained model have _different_ strengths, and that 3T can benefit by combining them, cf. our discussion of ensembling/distillation methods above.
> > > > The results below are for the zero-shot tasks of the IN-21k pre-trained L scale model setup of Table 3. (We are still working out some memory issues for the g-scale JFT results.)
> > > >
> > > >
> > > > For the scenario that you mention, where the baseline and LiT are similarly strong, we wish to highlight the results for EuroSat. In Table 3 from the paper, we observe accuracies of 32.7% for the baseline and 27.6% for LiT. Yet, 3T obtains 42.8% accuracy. This is only possible if, here, the combination of pre-trained model and contrastive learning in the 3T objective are better than either alone.
> > > >
> > > >
> > > > The first two columns of the table below show that there is predictive diversity between the Baseline and LiT.
> > > > The first column gives the proportion of datapoints where the Baseline but not LiT predicts on correctly.
> > > > The second column gives the proportion of datapoints where LiT but not the Baseline predicts on correctly.
> > > > We can see that, for both the Baseline and LiT, there is a significant proportion of inputs, where only one of the models predicts correctly.
> > > > Hence, the two models have different strengths and, equivalently, also make different mistakes.
> > > > A combination of the two approaches, such as 3T, can benefit from this.
> > > >
> > > >
> > > > In columns three, four, and five, we illustrate that 3T predicts differently from LiT and the Baseline.
> > > > The columns show the proportion of datapoints where 3T predicts differently than the baseline (3), LiT (4), or differently from both (5).
> > > > Clearly, 3T learns a novel predictive mechanism that is meaningfully different from LiT and the Baseline.
> > > > Whenever 3T outperforms both LiT and the Baseline (Caltech, DTD, IN-A, IN-R, ObjectNet, EuroSat, Sun397 in Table 3 in the paper) it must have learned to combine knowledge from the pre-trained model and contrastive learning mechanism in a beneficial way.
> > > >
> > > >
> > > > We hope you found this additional analysis insightful,  and would be highly interested to help clear up any other concerns that may stand in the way of you increasing your score further.
> > > >
> > > >
> > > > |                              |   Basel., Not LiT |  LiT, Not Basel. |   3T != Basel. |   3T != LiT |   3T != Basel. and 3T != LiT |
> > > > |:-----------------------------|------------:|-----------:|--------------:|-------------:|----------------:|
> > > > | IN-1k    |         6.9 |       13.3 |          22.8 |         26.5 |                   14.5 |
> > > > | CIFAR-100|         7.5 |       18.0 |          28.6 |         32.8 |                   20.4 |
> > > > | Caltech |         3.9 |        4.3 |          11.5 |         16.1 |                     8.4 |
> > > > | Pets    |         7.2 |        7.1 |          10.9 |         14.4 |                     5.5 |
> > > > | DTD     |        14.3 |        7.6 |          27.2 |         40.3 |                    19.5 |
> > > > | IN-A     |        19.0 |       11.7 |          38.3 |         52.8 |                   28.4 |
> > > > | IN-R     |        23.5 |        3.8 |          13.3 |         34.2 |                   10.0 |
> > > > | IN-v2    |         8.4 |       13.3 |          27.2 |         32.8 |                   18.0 |
> > > > | ObjectNet|        20.7 |        6.4 |          34.1 |         52.9 |                   26.7 |
> > > > | EuroSat |        19.2 |       14.2 |          62.3 |         75.1 |                    49.4 |
> > > > | Flowers |         4.0 |       14.8 |          25.4 |         27.0 |                    16.0 |
> > > > | RESISC  |        33.2 |        4.1 |          31.9 |         62.6 |                    25.7 |
> > > > | Sun397  |        11.8 |        9.6 |          21.9 |         31.2 |                    14.2 |

---

### Official Review · Reviewer_Wqsw · 2023-07-05

**Soundness:** 3 good
**Presentation:** 3 good
**Contribution:** 3 good
**Rating:** 6
**Confidence:** 3

**Summary:**

This paper presents Three Towers (3T), a method to enhance the contrastive learning of vision-language models by incorporating pretrained image classifiers. Unlike the previously proposed Locked-Image Text Tuning (LiT) method, which directly uses frozen pretrained classifier embeddings, 3T introduces a third tower containing the frozen pretrained embeddings and encourages alignment between the third tower and the main image-text towers. The paper shows that 3T consistently outperforms LiT and the CLIP/ALIGN baseline for retrieval tasks, and for ImageNet-21k-pretrained models, 3T also outperforms LiT for few- and zero-shot classification.

**Strengths:**

1. The paper presents a straightforward and effective approach to incorporating pretrained image models into contrastive learning of vision-language models.
2. The paper is well-written and easy to follow, with a clear explanation of the motivation, methodology, and experimental results.
3. The results demonstrate the effectiveness of the 3T method in improving the performance of vision-language models, particularly for retrieval tasks and few- and zero-shot classification with ImageNet-21k pretraining.
4. The additional training cost is limited, as the paper said "as frozen embeddings from the third tower can be pre-computed and then stored with the dataset"
5. The experiments in this paper are very solid and present extensive experiments on various tasks, pretraining datasets, and model sizes.

**Weaknesses:**

1 The reviewers perceive a resemblance between this work and SLIP[1], as both approaches combine CLIP with existing visual pretraining methods. While 3T employs a direct strategy using pretrained encoders, SLIP opts for a joint training approach. The reviewers recommend including a discussion and comparison between this paper and SLIP to provide a more comprehensive understanding.

2. This paper has less exploration of pretrained classifier embeddings and does not explore different pre-trained models as the third tower.
Experiments similar to SLIP[1] can be taken to explore the adaptation of visual coders with different pre-training methods to the CLIP framework.

3. The overall engineering style of the paper is heavy, lacking some insightful analysis and limited innovation.

[1] Mu, Norman, et al. "Slip: Self-supervision meets language-image pre-training." European Conference on Computer Vision. Cham: Springer Nature Switzerland, 2022.

**Questions:**

Have you ever attempted to calculate the loss function using the third tower at different layers? For instance, in a 12-layer model, you could compute the loss with the third tower in the hidden layer of layer 8 while calculating the regular loss in layer 12. This approach might offer valuable insights into the model's performance at various depths.

**Limitations:**

Addressed

---

> ### Author Rebuttal · Authors · 2023-08-09
>
> Dear reviewer Wqsw,
>
>
> Thank you for your hard work and helpful feedback. We will very gladly incorporate some of your excellent suggestions and look forward to engaging with you during the discussion period to clarify any remaining points. Please read our comment above, addressed to all reviewers, first.
>
>
> >The reviewers perceive a resemblance between this work and SLIP[1], as both approaches combine CLIP with existing visual pretraining methods. While 3T employs a direct strategy using pretrained encoders, SLIP opts for a joint training approach. The reviewers recommend including a discussion and comparison between this paper and SLIP to provide a more comprehensive understanding.
>
>
> Thanks for suggesting this! We were not aware of SLIP and agree it is related work. We have included a citation and discussion of SLIP in our updated version of the draft – note, we cannot currently update the submission on OpenReview. Below we give a detailed discussion of the relation between SLIP and 3T, which we believe is mostly _complementary_.
>
>
> With 3T, we study the benefits of incorporating _pre-trained image models_ into CLIP/ALIGN-style contrastive learning.
> The SLIP paper instead studies different ways of incorporating different _image-only self-supervision objectives_ into CLIP/ALIGN-style contrastive learning.
> With 3T, we assume there is a pre-trained model available.
> SLIP cannot include pre-trained models (except perhaps for initializing their encoders, i.e. the ‘fi’/’ft’ in Algorithm 1 of the SLIP paper).
> Therefore, SLIP is orthogonal to 3T: future work could _combine_ SLIP with 3T by replacing the loss between the main image and text tower in 3T with a SLIP loss, such that the main image tower benefits from the pre-trained model and SLIP objective.
> We note that, in addition to SLIP, there are many other approaches that improve CLIP (see related work) and that could benefit from pretrained models by combining them with a 3T-style approach.
>
>
> > This paper has less exploration of pretrained classifier embeddings and does not explore different pre-trained models as the third tower. Experiments similar to SLIP[1] can be taken to explore the adaptation of visual coders with different pre-training methods to the CLIP framework.
>
>
> Thanks for this great suggestion! We have performed experiments with self-supervised DINO models as well as supervised, but ResNet-based BiT models.
> Please see our reply to all reviewers (‘Additional Image Encoders’) for the results and discussion.
> In short, we find that 3T continues to outperform the baseline and LiT for retrieval tasks, and we are working towards providing results for additional models and tasks for the final submission.
>
>
> > The overall engineering style of the paper is heavy [...] and limited innovation.
>
>
> We agree that our proposal of the 3T architecture is largely straight-forward, although we do not think this is a bad thing at all: 3T is easy to implement and provides solid and consistent performance improvements.
> See, e.g., reviewer Y95u, who highlights this as a strength of our approach.
> Note the reviewer guidelines explicitly mention that even a ‘novel combination of well-known techniques’ can be ‘valuable’, although we believe our paper provides more than that.
>
>
> > lacking some insightful analysis
>
>
> Could you please clarify what you mean by ‘lacking some insightful analysis’?
> We believe our paper provides extensive empirical analysis of the 3T architecture, evaluating performance across many different downstream tasks, pre-trained models, and contrastive learning datasets (S. 4.1, S. 4.2, S. A.2.3).
> Further, we explore performance at multiple model scales (S. 4.3), the robustness of our method (S. 4.4), propose a novel interpolation approach between CLIP- and LiT-style prediction (S. 4.5, S. A.4), provide a thorough ablation (S. 4.6, A. 5), study performance at different training budgets (S. A.3), evaluate and interpret training dynamics (S. A.1), and study probabilistic robustness and out-of-distribution detection (S. A.2).
> All of these empirical results and ablations add valuable insight about the inner workings of the 3T method.
>
>
> > Have you ever attempted to calculate the loss function using the third tower at different layers? For instance, in a 12-layer model, you could compute the loss with the third tower in the hidden layer of layer 8 while calculating the regular loss in layer 12. This approach might offer valuable insights into the model's performance at various depths.
>
>
> Thanks for the suggestion! We have not tried this but agree that future work could consider this. It seems you might be suggesting that deeper layers implement more fine-grained features and that aligning to those might perhaps not be desirable. Are you aware of work that studies ViT losses/representations at different layer depths?
>
>
> More generally, our ablation study in S. 4.6 (and A.5) already investigates many different modifications to the 3T architecture, such as introducing weight hyperparameters or additional softmax temperatures.
> We find these do not give significant gains, and therefore favor the simplest version of 3T that works well.
> Computing losses between different layers would likely require that both image towers in 3T are comparable architectures, which reduces the flexibility of the method.
>
>
> Again, thank you very much for your review. We hope that our reply has addressed your concerns. Please let us know if there are any further changes you would like to see or if there is anything else that we can clarify.

---

> > ### Comment · Reviewer_Wqsw · 2023-08-20
> > **Re: Rebuttal by Authors**
> >
> > Overall, the reviewer is satisfied with the author's response.
> > The author is requested to include these results in the final version of the manuscript.
> > I have accordingly increased my rating.
> >
> > For some models, such as iGPT[1] ans MAE[2], when testing linear probe, the best performance is often not the last layer, but one of the middle layers.
> >
> > [1] Chen, Mark, et al. "Generative pretraining from pixels." International conference on machine learning. PMLR, 2020.
> > [2] He, Kaiming, et al. "Masked autoencoders are scalable vision learners." Proceedings of the IEEE/CVF conference on computer vision and pattern recognition. 2022.

---

> > > ### Author Response · Authors · 2023-08-20
> > > **Author Response**
> > >
> > > We are very glad to hear our rebuttal addressed your concerns and thank you for increasing your score!
> > >
> > >
> > > We will happily include the results we present here in the final version of the manuscript.
> > > Thank you for your interesting comment regarding iGPT and MAE.
> > >
> > >
> > > Please let us know if there is anything else that we can help clarify.

---

### Official Review · Reviewer_ni71 · 2023-07-05

**Soundness:** 3 good
**Presentation:** 4 excellent
**Contribution:** 3 good
**Rating:** 6
**Confidence:** 4

**Summary:**

In this paper, a technique called Three Towers (3T) is presented as a means to enhance the contrastive learning of vision-language models. The approach involves integrating pretrained image classifiers to achieve flexible and effective transfer in contrastive vision-language models. The authors provide a formalization of the 3T method, which serves as an extension to the LIT model by introducing an additional trainable Image tower.

**Strengths:**

- Based on the results shown in Tables 1, 2, and Table A.1.2, Three Towers (3T) demonstrates superior performance compared to both LIT and Baselines in retrieval tasks. Specifically, 3T outperforms LIT and Baselines in 2 out of 3 different experiments related to retrieval, indicating its effectiveness in this task.
- The paper can be considered as an experimental paper, as it presents a reasonable number of experiments and provides a thorough analysis of the results.
- While the contribution of the paper may not be groundbreaking, it explores a novel direction in contrastive learning of vision and language, which adds value to the existing research in the field.

**Weaknesses:**

One weakness of the Three Towers (3T) method is its performance in zero-shot and few-shot evaluations. While 3T outperforms LIT and baselines in the specific scenario of IN-21k pertaining to L scale models, Unfiltered WebLI, it underperforms LIT in other settings as reported in Table A.1 and A.2. This discrepancy is concerning, as it indicates that 3T excels only in 1 out of 4 different zero-shot and few-shot evaluations. Consequently, it is recommended that the authors modify the paper's title to accurately reflect this limitation, emphasizing the superior performance of 3T in retrieval tasks rather than zero- and few-shot tasks.

**Questions:**

- Regarding Line 214, the sentence raises a question about the caveat on LiT's few-shot evaluation. It is important to understand why there might be a caveat in this aspect of LiT's performance. According to my understanding, if the authors used the same data split as LIT for a few-shot evaluation, there should not be any caveat. Therefore, I request further elaboration on this matter.

- Grammar correction: Line 91 should be modified as follows: "then keep them frozen during training" (remove the second "them").

**Limitations:**

There are not any limitations.

---

> ### Author Rebuttal · Authors · 2023-08-03
>
> Dear reviewer ni71,
>
>
> Thank you for your hard work and helpful feedback. We look forward to actively engaging with you during the discussion period to clarify any remaining points. Please read our comment above, addressed to all reviewers, first.
>
>
> > One weakness of the Three Towers (3T) method is its performance in zero-shot and few-shot evaluations.
>
> We agree that 3T’s most positive results are achieved on retrieval tasks.
> However, we would like to push back on the claim that 3T is _only_ valuable for retrieval.
>
>
> Firstly, LiT’s few-shot classification performance is identical to the few-shot performance of the pretrained model and is not in any way improved during contrastive training – only methods like 3T can improve the few-shot classification performance of pre-trained image models using contrastive learning.
>
>
> Secondly, 3T outperforms LiT and the CLIP-style baseline when using IN-21k pre-trained models. JFT-pretrained models are not public, and 3T’s superior classification performance using public IN-21k pre-trained models will be relevant to a broad audience.
>
>
> Thirdly, even for JFT-pretrained models, LiT performance is sometimes worse than the baseline/3T for specialized tasks such as Eurosat and RESISC (Table A.2).
> (For IN-21k-based models there are more examples: DTD, IN-A, IN-R, ObjectNet, EuroSat, and RESISC, see Table 1.)
> We believe this is indicative of 3T being more robust than LiT, as its image tower can flexibly benefit from the diverse contrastive learning data _and_ knowledge in the pre-trained embeddings.
> Both the baseline and LiT sometimes perform much worse than all other methods – this is never the case for 3T.
> Our results in S. 4.4. with Places365 and the ‘mismatched’ setup further show 3T is more robust than LIT and the baseline.
>
>
> Further, S. 4.5 shows how 3T allows us to interpolate between LiT- and CLIP-style prediction, which boosts zero- and few-shot classification performance, and essentially allows one to to trade off their respective strengths and weaknesses at test time for a particular downstream task.
>
>
> In summary, we believe that, for classification tasks, 3T has three major selling points: (1) increased performance for IN-21k-pretrained models, (2) increased robustness across all setups, (3) flexibility in interpolating between LiT and 3T predictions.
> While we are happy to discuss changes to the paper title, we believe that any new title should account for these benefits of the 3T architecture.
>
>
> Lastly, we believe our detailed evaluation and open reporting of the classification performance of 3T – both its strengths and weaknesses – should not count against this submission.
> The performance of 3T on image classification does not take away from its performance on retrieval tasks: we do not believe it is necessary for a new method to always beat prior work (e.g. LiT) across all possible downstream tasks.
> Note, for example, that LiT itself usually does not improve over CLIP/ALIGN on retrieval tasks.
>
>
> > [...] This discrepancy is concerning, as it indicates that 3T excels only in 1 out of 4 different zero-shot and few-shot evaluations.
>
>
> We disagree with this summary of 3T’s classification performance.
> Of those 4 evaluations, only one pertains to IN-21k-pretraining; three use JFT-pretraining across different WebLI splits.
> We do not find large differences between the three WebLI splits in terms of classification performance in Table A.2.
> We therefore think it is best, and fairer, to summarize 3T classification performance as: (1) We do not find significant differences between models for different WebLi splits; (2) 3T outperforms LiT for IN-21k-pretrained models but not for JFT-pretrained models.
>
>
> To make sure of this, we are currently running experiments with IN-21k-pretrained models on the pair- and text-filtered WebLi splits, giving us a full set of WebLI splits for both IN-21k and JFT-pretrained models.
> Please see our reply to all reviewers (‘Additional WebLI Splits for IN-21k Pre-Training (Table R.3)) for these intermediate results. We will provide a full set of classification results as soon these runs are completed.
>
>
> However, as discussed above, even if 3T would only improve retrieval performance – noting it actually is significantly helpful in many classification scenarios – we do not believe it is necessary for a new method to always beat prior work across all possible downstream tasks (e.g. LiT does not improve retrieval compared to CLIP).
>
>
> > Regarding Line 214, the sentence raises a question about the caveat on LiT's few-shot evaluation. It is important to understand why there might be a caveat in this aspect of LiT's performance. According to my understanding, if the authors used the same data split as LIT for a few-shot evaluation, there should not be any caveat.
>
> Apologies, this is a misunderstanding and we will update the text in the revised version to clarify this point.
>
>
> Of course, we are using the same data split for all models for all downstream evaluations: the results are directly comparable.
>
>
> By ‘caveat’ we are referring to our discussion of LiT’s few-shot classification performance in l. 159.
> For few-shot classification evaluations, we train a linear classifier atop the features of the image tower (cf. l. 85).
> For LiT, these are the fixed embeddings from the pretrained model.
> Therefore, LiT’s few-shot performance does not change during contrastive training and is exactly the same as that of the pre-trained image model (see also Figure A.4).
> This is what we mean by 'caveat’.
> We apologize for the lack of clarity and will be explicit about this in the updated version of the draft.
>
>
> > Grammar correction …
>
>
> Thanks for spotting this! We will happily update the paper accordingly.
>
>
> Again, thank you very much for your review. We hope that our reply has addressed your concerns. Please let us know if there are any further changes you would like to see or if there is anything else that we can clarify.

---

> > ### Comment · Reviewer_ni71 · 2023-08-20
> > **Post-rebuttal comments**
> >
> > Thanks authors for their detailed clarifications and for providing the experiment I requested. I have also reviewed the comments made by other reviewers and the authors' responses to them. The authors' rebuttal effectively addressed my concerns, and as a result, I will not be lowering my score.

---

> > > ### Author Response · Authors · 2023-08-21
> > > **Author Response**
> > >
> > > Thank you for your reply!
> > > We are very glad to hear that we have effectively addressed your concerns.
> > >
> > > Please let us know if there is anything else we can help clarify, in particular, if it would lead you to consider increasing your score.

---

### Official Review · Reviewer_GeUR · 2023-07-07

**Soundness:** 3 good
**Presentation:** 3 good
**Contribution:** 2 fair
**Rating:** 5
**Confidence:** 4

**Summary:**

This paper focuses on improving LiT. A concern with LiT is that it may be overly reliant on the pretrained model, completely missing out on any potential beneﬁts the image tower might get from contrastive training. The proposed 3T method is formalized for ﬂexible and effective transfer of pretrained classiﬁers into contrastive vision-language models.

**Strengths:**

The proposed 3T method is formalized for ﬂexible and effective transfer of pretrained classiﬁers into contrastive vision-language models. 3T consistently improves over LiT and a from-scratch baseline for retrieval tasks. 3T also supports a simple post-hoc method to further improve performance by combining 3T- and LiT-like prediction.

**Weaknesses:**

1. Text-Encoder.
LiT showed that locked text tower performs badly, and finetuning gives small to negligible gains over training from scratch. But The 3T approach includes pretrained knowledge without suffering from the inﬂexibility of directly using the pretrained model as the main image tower. Maybe applying 3T-like architecture on the text tower can consistently solves the problem on text tower of LiT.

2. Dataset
In LiT, CC, YFCC datasets are used to train the whole model, while 3T uses WebLi. Why not uses exactly the same dataset as LiT? In that way I think the comparison would be more convincing.

3. Although 3T removes the reliance on the pretrained image tower, it brings more reliance on the constrastive-learning dataset. If the contrastive-learning is not diverse enough, the chances of encountering “blindspots” in downstream applications wound be higher.

4. Experiments with more pretrained models
LiT designed a control experiments with different pretrained models, i.e., MoCo, 	DINO, AugReg. I believe this experiment is also necessary for 3T.

5. Computational efficiency
One of the biggest adavantages of LiT is the computational efficiency, as it only optimizes the text encoder. In 3T, both image and text towers are trained during training, as well the additional cost of the pretrained image tower.

6. Generalization ability.
Lit has discussed the reason why locked is better than unlocked. The image and text towers will be aligned better if they are tuned together, but it will also make them specialized the dataset used for alignment. This is somewhat opposed to analysis in 3T.

**Questions:**

Please refer to the “Weakness”.

**Limitations:**

yes.

---

> ### Author Rebuttal · Authors · 2023-08-09
>
> Dear reviewer GeUR,
>
>
> Thank you for your hard work and helpful feedback. We will gladly incorporate some of your suggestions and look forward to engaging with you during the discussion period to clarify any remaining points. Please read our comment above, addressed to all reviewers, first.
>
>
> > 1 [...]
>
>
> Thanks for this great idea! We have now performed experiments with pre-trained language models. You can find the results in our reply to all reviewers. In short, unlike LiT, 3T does not suffer catastrophically when using pre-trained language models – however, it also does not seem to benefit from them.
>
>
> > 2 [...]
>
>
> Firstly, we would like to highlight that we always compare models that we have trained on the _same_ contrastive learning dataset. This ensures a fair comparison that avoids differences in contrastive learning datasets as a confounder.
>
>
> Secondly, in the LiT paper, Zhai et al. explore two different setups: once the small-scale CC/YFCC setup you mention and once a large-scale dataset of 4B image-text pairs, for which LiT performs better.
> There are no fundamental differences between the WebLI and large scale LiT data: both are noisy datasets of image-caption pairs.
> We simply prefer WebLI because it is newer and about 2.5 times larger for its largest split.
>
>
> Further, we presents results across three subsets of WebLI: the standard dataset, a pair-filtered split, and a text-filtered split (cf. S.4 and Table A.2).
> The standard WebLI in particular applies very similar (minimal) data filtering as that used in the LiT paper, and therefore, our results on this split should be comparable to those of Zhai et al.
> Our experiments across the three WebLI splits should therefore be fair, representative, and not underestimate LiT performance.
>
>
> However, to make sure, we are currently running additional experiments on the _exact same_ data that Zhai et al. use to train LiT.
> In Figure R.1 in the rebuttal pdf, we provide training curves of the ongoing runs.
> So far, we observe no significant difference to our WebLI runs.
> We will provide final numbers across all tasks as soon as possible.
>
>
> > 3 [...]
>
>
> Thanks for this suggestion. We will happily extend our discussion on 'Risk in Locking’ to include benefits of locking, too, as soon as we can update the paper.
>
>
> We would generally expect contrastive learning datasets to have significantly fewer blindspots than classification datasets, as they contain a wide variety of web-scraped data, are much larger, and do not rely on manual annotation.
> It is true that, compared to LiT (but not the baseline), 3T increases dependence on the contrastive learning data.
> We believe this is usually a good thing and not a weakness: unlike LiT and the baseline, 3T makes use of two sources of information for learning the image tower.
> To some extent, this allows 3T to guard against ‘blindspots’ of the pre-trained model _and_ the contrastive learning data.
> E.g., Stanford Cars is a blindspot in IN-21k models, but 3T does not suffer from the same failure as LiT and even manages to improve over the baseline (cf. Table 1 here and in the following).
> Conversely, it seems that the Birds task could be considered a blindspot of the contrastive learning data: it requires fine-grained knowledge of bird species, which is heavily present in IN-21k and presumably more so than in our contrastive learning data.
> On Birds, the baseline performs worse than LiT by 30 %p., but 3T significantly closes the gap to about 18 %p.
>
>
> Therefore, in terms of robustness to blindspots – or just overall robustness, cf. S. 4 – 3T is clearly advantageous to both LiT and the CLIP-style baseline. Unlike 3T, both LiT and the baseline sometimes perform significantly worse than all other approaches, but this never happens for 3T.
>
>
> > 4 [...]
>
>
> Thanks for this great suggestion! We have performed experiments with self-supervised DINO models as well as supervised, but ResNet-based BiT models.
> Please see our reply to all reviewers (‘Additional Image Encoders’) for results and discussion.
> In short, we find that 3T continues to outperform the baseline and LiT for retrieval tasks, and we are working towards providing results for additional models and tasks for the final submission.
>
>
> > 5 [...]
>
>
> We agree increased training cost is a weakness of the 3T architecture over LiT.
> However:
> (1) We acknowledge increased training costs clearly as a limitation in S. 6;
> (2) As benefits for the increased training costs, 3T obtains better retrieval performance and increased robustness;
> (3) Inference costs are the same for all methods. These can practically be much more important than training costs if a model is trained once and then deployed many times (see e.g. Patterson et al., 2022, https://arxiv.org/abs/2204.05149).
> Therefore, we feel that increased training costs should not hold back acceptance of this paper.
>
>
> > 6 [...]
>
>
> Apologies, but we are not quite sure what exactly you are referring to here. It would be great if you could clarify where exactly you think our analysis opposes conclusions of Zhai et al. (2022).
>
>
> Are you saying that Zhai et al. claim that, in CLIP, the image and text towers are better aligned on contrastive data than for LiT, but this does not transfer to better downstream task performance?
> We do not oppose, but rather fully agree with this: the contrastive training loss is often not indicative of downstream task performance. We also observe this in Appendix A.
> We agree with Zhai et al. that there are benefits to using pre-trained models in contrastive learning but prefer a more flexible approach, which achieves better retrieval performance (where LiT’s fixed embeddings are perhaps not fine-grained enough) and higher robustness (where LiT’s embeddings may have blindspots).
>
>
> Again, thank you very much for your review. We hope that our reply has addressed your concerns. Please let us know if there are any further changes you would like to see or if there is anything else that we can clarify.

---

> > ### Author Response · Authors · 2023-08-20
> > **Discussion Period Almost Over**
> >
> > Dear Reviewer GeUR,
> >
> >
> > The discussion period ends in less than 28 hours.
> >
> > We have made significant efforts to address your concerns in our rebuttal, which we believe has significantly strengthened our submission.
> >
> > We would greatly appreciate it, if you could engage with us and our rebuttal during the last remaining hours of the discussion period.
> >
> >
> > Best wishes
> >
> > The Authors

---

> > ### Comment · Reviewer_GeUR · 2023-08-21
> > **Post-rebuttal comments**
> >
> > Thank the authors for their clarifications and address all my concerns. I would like to raise my score to borderline accept.

---

> > > ### Author Response · Authors · 2023-08-21
> > > **Author Response**
> > >
> > > Thank you for your reply. We are very glad to hear our clarifications have addressed all your concerns.
> > >
> > > Please let us know if there is anything else we can do or help clarify.

---

### Author Rebuttal · Authors · 2023-08-09

We thank all reviewers for their careful consideration, insightful comments, and helpful suggestions. We are glad to see the paper was generally well received and hope that our responses alleviate the concerns that were raised. We believe your input has already helped improve the paper and look forward to engaging with you further during the discussion period.


We were glad to see all reviewers recognize the empirical improvements of 3T, highlighting that ‘3T consistently improves over LiT and a from-scratch baseline for retrieval tasks’ (GeUR), ‘demonstrates superior performance compared to both LIT and Baselines in retrieval tasks’ (ni71), ‘the results demonstrate the effectiveness of the 3T method in improving the performance of vision-language models, particularly for retrieval tasks and few- and zero-shot classification with ImageNet-21k pretraining' (Wqsw), and 'empirical gains are good, especially in retrieval tasks’ (Y95u).


Further, we were glad to see you felt that 3T 'explores a novel direction [...] which adds value to the existing research in the field’ (ni71), that our presentation is 'excellent ‘ (ni71, Y95u), that the paper is 'well-written and easy to follow, with a clear explanation of the motivation, methodology, and experimental results’ (Wqsq), and that our idea is 'well described’ (Y95u). Lastly, we were glad to see you recognize our empirical evaluation as ‘thorough’ (ni71), 'very solid’/'extensive’ (Wqsw), highlighting that we perform 'proper ablations’ (Y95u).


Following your excellent feedback, we have added a number of new experiments, which are attached to this message as a PDF. Note that, as per NeurIPS guidelines, we cannot update the paper directly during the rebuttal. We will include these results in the final submission. However, many of our experiments require a large number of accelerators and multiple days of compute which makes it difficult to finish them within the time constraints of the rebuttal period.
We therefore sometimes need to present intermediate results for ongoing experiments.
Note, this early comparison generally benefits LiT as the benefits of a locked image tower are strongest early on in training, cf. Figures A.1 (c) and A.4.
For the baseline, the gap to 3T is usually consistent throughout training.
Therefore, our intermediate results here should be representative for comparing 3T and the baseline and conservative for comparing 3T and LiT.
We will provide a final and full set of results as soon as possible.


Concretely, we study the following new experiments:


* **Additional Image Encoders** (Table R.1): We study additional pre-trained image encoders: a ResNet-based BiT model trained on IN-21k (Kolesnikov et al., ECCV 2020, arXiv: 1912.11370) and a self-supervised encoder based on DINO as in Zhai et al. (2022). We find that 3T continues to outperform the baseline and LiT for retrieval tasks for both encoders. We are currently working on results for additional image encoders, such as MoCo-v3 and AugReg as in Zhai et al. (2022), as well as a full set of classification evaluations.

* **Pre-Trained Language Models** (Table R.2): We explore if there are benefits to using pre-trained _language models_ instead of image encoders with 3T. For LiT, we confirm the results of Zhai et al. (2022) that performance suffers drastically when using a locked pre-trained language encoder as the text tower. In contrast, for 3T, we do not observe a drastic decrease in performance, once more showing that 3T is more robust to deficiencies in the pre-trained model than LiT. However, we also do not observe gains from using a pre-trained language model with 3T, justifying our choice of focusing on pre-trained image encoders in the original submission. Nevertheless, we think it is plausible that future work may yet find benefits of incorporating knowledge from pretrained language models in other settings or downstream tasks, e.g. tasks that require more complex language reasoning abilities.

* **Additional WebLI Splits for IN-21k Pre-Training** (Table R.3): We study IN-21k pre-trained models at L scale for the text-filtered and pair-filtered splits of the WebLI dataset (see S. 4). We find that 3T continues to outperform the baseline and LiT for retrieval tasks on these additional WebLi subsets.


* **Results on the Original LiT Dataset** (Figure R.1): We have started experiments for the baseline, LiT, and 3T on the exact same dataset Zhai et al. (2022) use to train LiT. While runs are still ongoing, so far, we observe no significant differences in behavior to our previous experiments on splits of the WebLI dataset.


We address your specific comments in the individual replies below.
We look forward to engaging with you during the discussion period to clear up any remaining concerns and allow us to improve the paper further.

---

> ### Author Response · Authors · 2023-08-18
> **Update: Final Results for 'Additional Image Encoders'**
>
> Dear reviewers,
>
>
> We are glad to see that some of you have already responded (quite positively) to our rebuttal, and we hope to engage with those that have not yet responded in the last remaining days of the discussion period.
>
>
> We are now able to provide final results for the experiments with ‘Additional Image Encoders’ (Table R.1 in the rebuttal pdf).
> Again, we would like to thank you for suggesting these experiments: as we detail below, 3T performs quite well for DINO- and BiT-based pre-trained models on retrieval _and_ classification tasks.
> We also hope for your understanding that our large scale experiments require a significant number of accelerators and multiple days of compute, such that we had to provide intermediate results last week at the time of the rebuttal deadline.
>
>
> As per NeurIPS guidelines, we are unable to update the rebuttal pdf, and so we provide the updated results in the table below.
> Our conclusions for the final results exceed our expectations from the intermediate results.
> Concretely:
>
> * 3T consistently outperforms the baseline and LiT for DINO- and BiT-based pretrained models for all retrieval scenarios: for COCO and FLICKR tasks for the img2txt and txt2img direction.
>
> * For few-shot classification, LiT performs best on average for BiT based models but 3T performs best on average for our DINO experiments.
>
> * For zero-shot classification, 3T performs best on average for both our DINO- and BiT-based experiments.
>
>
> Again, we thank you for suggesting these experiments, as we believe they significantly strengthen our submission.
> |             |   Baseline |   (LiT, BiT)  |   (3T, BiT) |   Baseline | (LiT, DINO)  |   (3T, DINO) |
> |:-------------------|-----------:|-------:|-----:|-----:|-------:|-----:|
> | Ret Flickr img2txt |       71.2 |   61.1 | **74.6** |       71.2 |   60.4 | **74.0** |
> | Ret Flickr txt2img |       51.3 |   42.9 | **54.5** |       51.3 |   38.3 | **54.0** |
> | Ret COCO img2txt   |       44.3 |   40.9 | **46.8** |       44.3 |   38.9 | **45.7** |
> | Ret COCO txt2img   |       29.0 |   24.7 | **32.0** |       29.0 |   21.2 | **31.4** |
> | **Retrieval Average**| 48.9| 42.4| **52.0**|  48.9| 39.7| **51.3**|
> | FS Caltech         |       89.8 |   90.4 | **90.9** |       89.8 |   91.0 | **92.1** |
> | FS UC Merced       |       89.3 |   **91.6** | 91.2 |       89.3 |   **94.2** | 92.1 |
> | FS Cars            |       75.0 |   42.4 | **77.1** |       75.0 |   49.4 | **78.7** |
> | FS DTD             |       66.5 |   66.8 | **69.4** |       66.5 |   63.5 | **70.1** |
> | FS Col-Hist        |       68.3 |   **84.8** | 72.9 |       68.3 |   **85.6** | 75.2 |
> | FS Birds           |       44.5 |   **85.5** | 53.2 |       44.5 |   **62.1** | 52.5 |
> | FS Pets            |       75.8 |   **89.2** | 78.9 |       75.8 |   **82.5** | 76.3 |
> | FS IN-1k           |       52.0 |   **73.8** | 56.2 |       52.0 |   **60.4** | 56.4 |
> | FS CIFAR-100       |       57.8 |   **74.1** | 62.1 |       57.8 |   **62.3** | 61.0 |
> | **Few-Shot Average** | 68.8| **77.6**| 72.4| 68.8|  72.3| **72.7**|
> | 0S Pets            |       75.4 |   **80.9** | 80.2 |       75.4 |   **83.5** | 78.7 |
> | 0S IN-1k           |       59.8 |   **67.1** | 62.8 |       59.8 |   **62.9** | 62.5 |
> | 0S CIFAR-100       |       60.0 |   **72.6** | 59.6 |       60.0 |   56.0 | **61.1** |
> | 0S IN-A            |       32.8 |   26.2 | **34.9** |       **32.8** |   21.7 | 32.5 |
> | 0S IN-R            |       75.4 |   55.4 | **78.1** |       75.4 |   57.0 | **77.4** |
> | 0S IN-v2           |       53.1 |   **58.8** | 55.7 |       53.1 |   55.0 | **55.6** |
> | 0S Objectnet       |       44.7 |   34.4 | **47.3** |       44.7 |   25.3 | **45.0** |
> | 0S Caltech         |       79.1 |   76.7 | **79.3** |       79.1 |   79.0 | **80.7** |
> | 0S DTD             |       51.4 |   46.6 | **54.1** |       51.4 |   41.9 | **52.4** |
> | 0S Eurosat         |       **36.5** |   28.1 | 32.6 |       **36.5** |   32.6 | 36.2 |
> | 0S Flowers         |       51.9 |   **64.1** | 57.1 |       51.9 |   48.2 | **55.6** |
> | 0S Resisc          |       **53.1** |   27.8 | 52.9 |       **53.1** |   24.4 | **48.0** |
> | 0S Sun397          |       62.6 |   60.9 | **65.3** |       62.6 |   53.7 | **63.2** |
> | **Zero-Shot Average** |56.6 | 53.8 | **58.5**|56.6| 49.3| **57.6**|
>
>
> [edit]: Add Caltech and DTD zero-shot tasks; 3T performs best for both.

---

### Decision · Program_Chairs · 2023-09-21

**Decision:**

Accept (poster)

**Comment:**

This paper introduces an innovative technique known as Three Towers (3T) aimed at enhancing the contrastive learning of vision-language models by integrating pretrained image classifiers. In contrast to the LiT method, the 3T approach acts as an extension to the LIT model, introducing an additional trainable Image tower to promote alignment between the third tower and the primary image-text towers. The results demonstrate that 3T has led to notable improvements in retrieval tasks and few-/zero-shot classification.

Reviewers have praised the simplicity and clarity of the proposed 3T method, which is well-documented and supported by robust experimental evidence. The authors' responses to reviewers' concerns, including the exploration of additional image encoders, elucidating how 3T enhances predictions over LiT and the baseline, its applicability to Pre-Trained Language Models, essential comparisons with SLIP, considerations of pretraining scale, and detailed clarifications regarding fair comparisons with LiT and performance in zero-/few-shot evaluation, have all been meticulously addressed.

Given these substantial enhancements, all reviewers have expressed scores surpassing the acceptance threshold, and AC concurs with their recommendation to accept this paper. AC strongly encourages the authors to integrate these valuable insights, extensive experiments and constructive feedback from the reviewers into the final version of the manuscript.